# The effect of wildfires on flood risk: A multi-hazard flood risk approach for Ebro River basin Spain

Samuel Jonson Sutanto[1], Matthijs Janssen[1], Mariana Madruga de Brito[2], and Maria del Pozo Garcia[1]

[1]Earth Systems and Global Change Group, Wageningen University and Research, Wageningen, The Netherlands
[2]Department of Urban and Environmental Sociology, Helmholtz Centre for Environmental Research - UFZ, Leipzig, Germany

**Correspondence:** Samuel Sutanto. Earth Systems and Global Change Group, Wageningen University and Research, P.O. Box 47, 6700 AA, Wageningen, The Netherlands (Email: samuel.sutanto@wur.nl)

**Abstract.** Climate change increases the risk of wildfires and floods in the Mediterranean region. Yet, wildfire hazards are often overlooked in flood risk assessments and treated in isolation, despite their potential to amplify floods. Indeed, by altering the hydrological response of burnt areas, wildfires can lead to increased runoff and amplifying effects. This study aims to comprehensively assess flood risk using a multi-hazard approach, considering the effect of wildfires on flood risk, and integrating diverse socio-economic indicators with hydrological properties. More specifically, this study investigates current and future flood risks in the Ebro River basin in Spain for the year 2100 under the Shared Socioeconomic Pathways (SPP) 1-2.6 and SSP5-8.5 scenarios, taking into account projected socio-economic conditions and the effect of wildfires. An Analytical Hierarchy Process (AHP) approach is employed to assign weights to various indicators and components of flood risk based on insights gathered from interviews with seven experts specializing in natural hazards. Results show that the influence of wildfires on baseline flood risk is not apparent. Under the SSP1-2.6 scenario, regions with high flood risk are expected to experience a slight risk reduction, regardless of the presence of wildfires, due to an expected substantial development in adaptive capacity. The highest flood risk, almost double compared to the baseline, is projected to occur in the SSP5-8.5 scenario, especially when considering the effect of wildfires. Therefore, this research highlights the importance of adopting a multi-hazard risk management approach, as reliance solely on single-risk analyses may lead to underestimating the compound and cascading effects of multi-hazards.

## 1 Introduction

Floods, among all natural hazards, are known as the most frequently occurring natural hazard with immense adverse impacts on both society and the environment (Cai et al., 2019). The likelihood and severity of flooding are expected to increase even further in the future, driven by climate change, land-use change, and population growth (Jongman et al., 2012; Rentschler et al., 2023). Climate change not only enhances the frequency and intensity of heavy precipitation events but also increases the duration and occurrence of dry spells and heatwaves, leading to droughts and water shortages that, in turn, can trigger wildfires (Saaroni et al., 2015; Mazdiyasni and AghaKouchak, 2015; He et al., 2022). As such, floods and wildfires are interlinked hazardous events since wildfires can potentially create more vulnerable conditions to subsequent flooding, thereby increasing flood probability and severity (Versini et al., 2013). For instance, wildfire occurrences leave soils vulnerable to surface runoff

and erosion due to the loss of vegetation and alteration of soil properties (Moftakhari and AghaKouchak, 2019). The occurrence of flooding after wildfires poses challenges to societies, environmental ecosystems, and ecological systems. Understanding and assessing the impacts of these natural hazards is thus of utmost importance to prepare for and mitigate the consequences of floods and wildfires in the years ahead (Lehner et al., 2006).

Besides the changing climatic conditions, a better understanding of the socio-economic drivers of flood disasters is needed to
30 address their risks. The global population's rise has resulted in an increased number of people exposed to disasters (Rentschler et al., 2023). Astonishingly, more than a quarter of the global population has already been impacted by flooding over the last 20 years, not only physically but also economically (Tabari et al., 2021). Urbanization and land-use changes have also augmented society's exposure and vulnerability to floods, thereby escalating expected losses due to the increasing economic development (Tabari et al., 2021). Moreover, income inequalities contribute to the diversification of the impacts of flooding. Research has
35 shown that economically disadvantaged people are impacted the most as they possess fewer resources and capabilities to adapt to and prepare for floods (Brouwer et al., 2007). Therefore, it is crucial to enhance the economic resilience and capacity of the population to effectively prepare for, cope with, and recover from disasters. At the same time, institutional capacity is equally pivotal in reducing citizens' vulnerability (Romero-Lankao et al., 2013). Hence, besides hazard aspects, both economic and institutional capacities should be considered in risk assessments.

Analyzing future flood risk, considering changes in climatic conditions, socio-economic drivers, and the effects of wildfires on flood risk, presents a multifaceted and intricate challenge. The complexity stems from three primary sources. First, there is the inherent uncertainty of the likelihood of wildfire or flooding occurrence, denoted as the hazard probability. Second, the hazard exposure and the vulnerability of social and economic systems are dynamic and rapidly evolving (Klijn et al., 2015; Sword-Daniels et al., 2018; Moreira et al., 2021). Third, natural hazards often yield multiple interconnected effects that lead to
consecutive disasters (de Ruiter et al., 2020; de Brito, 2021). Concerning the third source of complexity, despite amplifying the risk of floods, the effects induced by wildfires are often given little consideration in conventional flood assessments (Versini et al., 2013). Exceptions include studies such as Versini et al. (2013), which assessed flood occurrence in the Ebro river basin, Spain, after wildfires and projected future flood probability. However, their study primarily focused on the hydrological probability of flooding, overlooking key socio-economic indicators and land-use changes anticipated. These indicators are,
however, key to evaluating flood risk, as they address the exposure and vulnerability of societies to floods, transcending the mere measurement of hydrological flood probability (Ologunorisa, 2004; Moreira et al., 2021).

Building upon the prior research conducted by Versini et al. (2013), we adopted a comprehensive approach to assess current and future flood risk in the Ebro River basin. We consider the effects of wildfires and other socio-economic indicators, such as population density and GDP. In this study, we investigate how wildfires indirectly trigger or amplify the risk of flooding
by changing soil characteristics (Tilloy et al., 2019; de Ruiter et al., 2020; De Angeli et al., 2022). Integrating wildfires and socio-economic indicators is crucial for identifying hotspot areas prone to flood risk, which has not been done before. Therefore, this study goes beyond conventional assessments by proposing advancements in three directions. First, we evaluate the interaction between wildfires and floods regarding disposition alteration (Tilloy et al., 2019; De Angeli et al., 2022). Second, we incorporate diverse socio-economic indicators into the flood risk assessment, considering both exposure and vulnerability

dimensions. For the exposure component, we include variables such as population, economic values of the regions, and road infrastructure. For the vulnerability component, we consider physical and social factors, such as topography, land cover, soil infiltration capacity, economic capacity, and institutional capacity (see Chapter 2.2). The use of these wide-ranging exposure and vulnerability indicators is a novel aspect of our study, which has not been included in many studies (e.g., Brouwer et al., 2007; Foudi et al., 2015; Gain et al., 2015; Cai et al., 2019). Lastly, we incorporate spatial and temporal dynamics by projecting future flood risk. We compare this historical flood risk with the Shared Socioeconomic Pathways (SSPs) and Representative Concentration Pathways (RCPs) projections for the year 2100 under the climate scenarios SSP1-2.6 and SSP5-8.5. Section 2 provides an in-depth exploration of the methods and data employed in the flood risk assessment. Chapter 3 presents the maps delineating each component of flood risk, including detailed exposure and vulnerabilities, and flood risk assessment for both the present and future. We discuss the findings in Chapter 4 and draw the conclusion in Chapter 5.

## 2 Methods and data

### 2.1 The Ebro River basin

The Ebro River basin is a major river basin in the northern region of Spain (Fig. 1). The river spans approximately 928 km, with a drainage area of 85,500 km$^2$ (Silva et al., 2011). Originating at an elevation of around 2,000 m.a.s.l in the Cantabrian mountains, the river flows from the northwest to the southeast, ultimately draining into the Mediterranean Sea between the cities of Barcelona and Valencia (Almazán-Gómez et al., 2019; Romaní et al., 2011). The Ebro River basin can be divided into three sub-basins: the Upper Ebro, extending from Cantabria (limited by the Iberian range and the Pyrenees) to Miranda de Ebro; the Middle Ebro, representing the largest sub-basin from Haro to Mequinenza; and the Lower Ebro, measuring 115 km in length, which serves as the confluence point for tributaries of the Ebro originating from the Cinca-Serge system to the delta into the Mediterranean Sea (Balasch et al., 2019).

Climate and hydrology in the Ebro basin differ significantly across its three sub-basins. Overall, the climate is Mediterranean, with some continental characteristics and a semi-arid climate in the central part of the basin. On average, the annual precipitation was estimated to be 622 mm, averaged from 1920 to 2000 (Balasch et al., 2019). The Upper Ebro experiences milder temperatures and higher precipitation, ranging between 1,000-1,500 mm annually. In the Middle Ebro, the average precipitation is lower, varying from 400 to 700 mm annually. Last, the Lower Ebro receives less than 400 mm (Balasch et al., 2019). The Upper Ebro hydrological regime highly relies on snowfall and snow retention. In contrast, the Middle and Lower Ebro are rainfall-driven basins, with peak flow occurring in spring and autumn and a discernable reduction during summer. Furthermore, the hydrological regime is significantly influenced by the several dams constructed throughout the basin. These dams are pivotal in regulating the river's flow and hydrological dynamics.

The Ebro basin is home to approximately 2.8 million people, accounting for 7,3% of Spain's total population (Almazán-Gómez et al., 2019). Among its major cities, Zaragoza and Pamplona are the biggest ones. The average population density of the basin is 38 people/km$^2$ (Silva et al., 2011; Terrado et al., 2006). Other notable cities with populations exceeding 100,000 inhabitants are Lleida, Logroño, and Vitoria-Gasteiz. Nearly 40% of the entire basin is sparsely inhabited, with fewer than 5

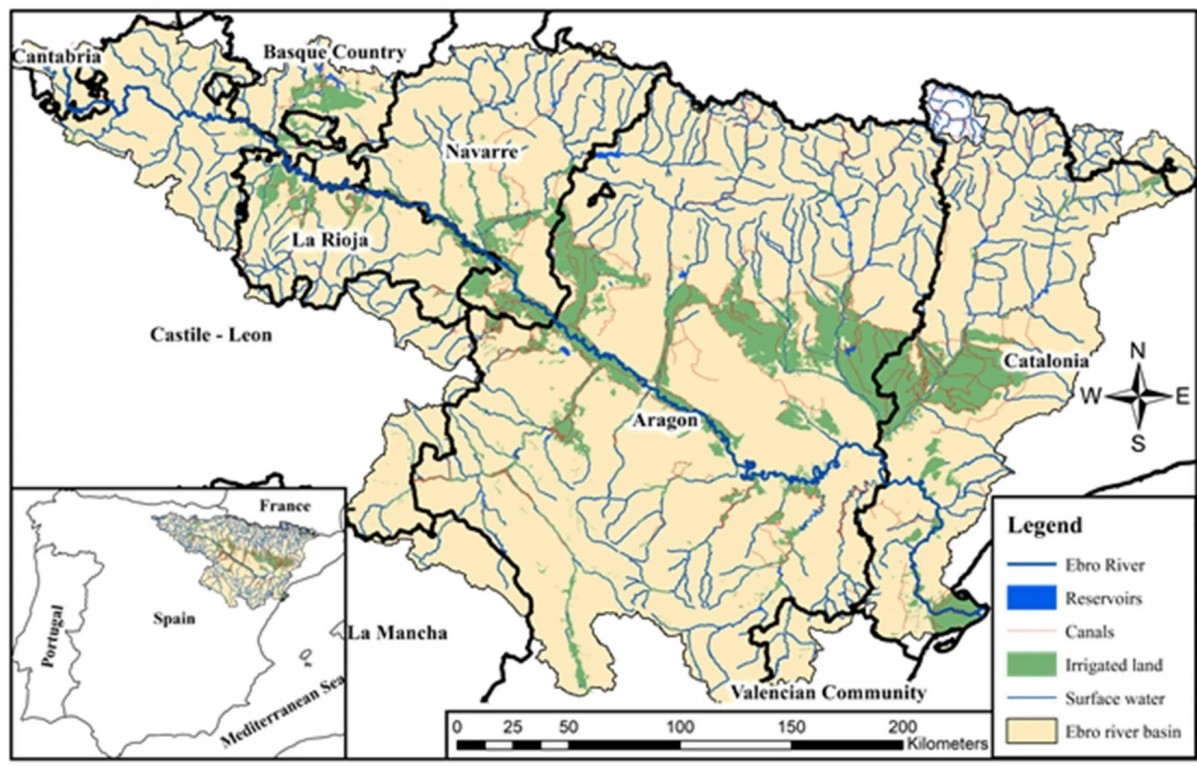

**Figure 1.** Basic hydrologic information of the Ebro River basin. Derived from Almazán-Gómez et al. (2021).

people/km$^2$, and is therefore considered uninhabited (Romaní et al., 2011). The total Gross Domestic Product (GDP) of the Ebro River basin contributes significantly to Spain's economy, comprising 8.5% of the nation's total GDP (Almazán-Gómez

et al., 2019), which accounts for approximately 102.6 billion euros in the year 2021 (Statista, 2022). Zaragoza and Pamplona's regions serve as economic hubs within the basin, hosting diverse economic activities, such as industry and agriculture (Grantham et al., 2013).

On the institutional level, the Hydrographical Ebro Confederation ("*Confederación Hidrográfica del Ebro*") holds a significant authority within the basin, overseeing various plans related to flood management in alignment with the European Water

Framework Directive (Romaní et al., 2011). In contrast, wildfire management is addressed at different spatial levels, rather than basin-wide coordination. This discrepancy is also evident at the European level, where the European Flood Directive (2007/60/EC of the European Parliament) focuses on assessing and managing flood risk. However, there are only policies aimed at protecting the EU's forests against fire, which have not been translated into a European fire directive. Therefore, flood management appears to have a higher priority than fire management due to higher civil protection standards associated with

flooding (Grantham et al., 2013).

## 2.2 Indicators for flood risk assessment, including wildfire risk

We conducted a comprehensive flood risk assessment by estimating both the probability of flood and wildfire hazards in the context of multi-hazard risk framework and the potential consequences of the flooding event under specific socio-economic contexts. To this end, we integrated these various elements using GIS (ArcGIS Pro), going beyond hazards to consider critical contextual socio-economic and geophysical indicators, included within exposure and vulnerability components. These indicators comprise e.g., institutional capacity, economic values, land use and land cover, geographic conditions, population, and the available infrastructures (Merz et al., 2014), as shown in Figure 2. These indicators were selected based on previous literature reviews (Supplementary Information Tables S1 and S2 provide detailed references for data). Data availability constraints limited a broader selection of indicators. Nevertheless, we employed a variety of indicators that allowed us to gain a more holistic understanding of flood risk.

To incorporate the effect of wildfires into our analysis, we considered both wildfire and flood hazard maps. The burnt area data indicates the wildfire hazard for the baseline scenario, while the Fire Weather Index (FWI) indicates the probability of fire danger prediction (Van Wagner, 1987; Abatzoglou et al., 2019). The FWI is a fire danger index derived from meteorological data that accounts for the effects of fuel moisture and weather conditions on fire risk. These meteorological data include temperature, relative humidity, wind speed, and precipitation. The FWI has been used worldwide to identify the potential for fire occurrence (de Groot and Flannigan, 2014; Field et al., 2015) and to estimate future fire activity under different climate scenarios (Grantham et al., 2013; Quilcaille et al., 2023). Thus, this study also utilizes the FWI to project future fire occurrences.

The recovery time of vegetation can vary significantly, ranging from a few years to several decades, depending on the ecosystem types and fire severity (Petropoulos et al., 2014). Gimeno-García et al. (2007) show that runoff is only slightly higher in burnt areas than in unburnt areas after 8 years. Therefore, this study assumed a recovery time of 8 years for vegetation and ecosystems for the baseline scenario, taking values from burnt areas between 2010 and 2018. However, for the year 2100, the recovery time of vegetation is neglected, as the FWI values are considered. One should note that while FWI indicates the likelihood of fires derived from climatic indices, burnt areas represent observed data derived from satellite images. Since current and future flood hazard maps are not available for the study area, we elaborated on them by considering the methodology by Seibert et al. (2010). Therefore, our analysis used the runoff coefficient ($K$) as a key metric for quantifying the amount of runoff relative to the volume of precipitation in the basin. A higher runoff coefficient signifies an increased likelihood of floods.

The exposure component consists of the population exposed to the hazard. It also includes economic values, quantified as the total GDP per region, and exposure related to road infrastructure, measured in terms of distance from roads/highways to flood zones (Fig. 2). The total GDP per province reflects the exposure concerning economic damages. However, most provinces are not fully included in the Ebro River basin, as the basin is unevenly distributed over the provinces. Therefore, a weighted average is used to adjust for the uneven distribution. To calculate the weighted average total GDP, the surface area in $m^2$ was calculated by ArcGIS Pro. This value was then divided by the total surface area of each province and multiplied by the total GDP for that province. Furthermore, we incorporated the distance from the rivers to reflect the level of exposure to the flood hazard. The closer an element is situated to the river, the higher its exposure to potential flooding (Zhang et al., 2020). This was determined

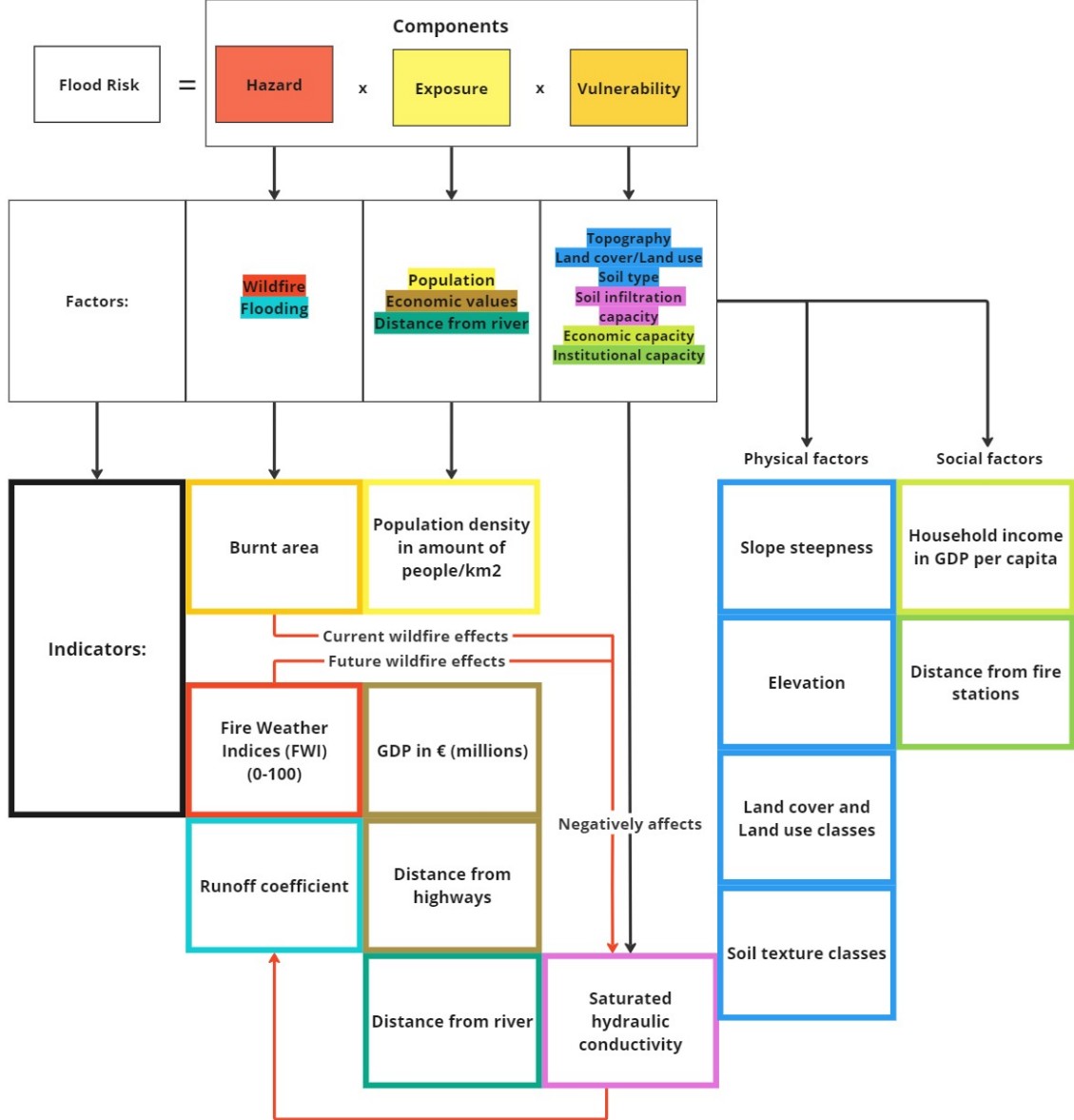

**Figure 2.** Overview of components and indicators considered in assessing the flood risk. For clarity, each component of risk is represented by different colors. The multi-hazard mechanism is illustrated by red arrows, indicating that wildfires, represented by burnt area and Fire Weather Index (FWI), alter the saturated hydraulic conductivity. This alteration leads to a higher runoff coefficient, thereby increasing the risk of flooding.

by arbitrarily selecting the main streams with cumulative lengths of 50,000 meters or more. Only highways were selected as they play a major role in transportation. Provincial and local roads were not considered as part of the exposure components. To

calculate the distance from highways, we considered the Euclidean distance. The risk classification for distance to highways and rivers was adapted from Roy et al. (2021).

The physical vulnerability component was considered to be linked to the effects of wildfires on the soil's infiltration capacity, measured as saturated hydraulic conductivity (Versini et al., 2013). Wildfires transform vegetated areas into bare soil, resulting in a higher vulnerability index to flooding due to reduced infiltration capacity. Physical vulnerability is also influenced by several other indicators, such as topography, measured in slope steepness and elevation, land cover/land use, and soil texture. The elevation indicates how vulnerable areas are to flooding, with lower-lying areas being more susceptible to inundation and, thus, flooding risk. For the slope steepness, steeper slopes increase flow velocity, influencing vulnerability to flooding (Rahmati et al., 2016). The soil texture indicates the infiltration rate of water moving through the soil. The higher the infiltration rate, the lower the runoff, and vice versa (Berhanu et al., 2013). The saturated hydraulic conductivity ($Ksat$), measured at a soil depth of 15 cm, reflects soil's ability to infiltrate water when saturated (Mohanty et al., 1994). The $Ksat$ was estimated with and without considering the wildfire effects. Nonetheless, the social vulnerability can be mitigated by the society's adaptive capacity. Adaptive capacity comprises a range of indicators, such as awareness, training, and available technology. In this study, due to the complexity of assessing these drivers, we focused on two key social vulnerability indicators: economic capacity and institutional capacity (Thanvisitthpon et al., 2020). Economic capacity, such as household income measured in GDP, serves as a measure to assess the community's ability to prevent, cope with, or recover from wildfires and floods. Higher GDP per capita values indicate a greater economic capacity to mitigate the negative effects of flooding, resulting in lower vulnerability. Additionally, institutional capacity, represented by indicators like the number of fire stations in the area dedicated to protecting the population from wildfires and floods, measures the capability of institutions to assist in preventing, managing, and recovering from these natural disasters (McLennan and Birch, 2005). This study focuses solely on the number of fire stations, as data on personnel and budget are not freely available and are difficult to quantify for future scenarios. Areas closer to fire stations are considered to have a lower vulnerability to flooding due to quicker response times in case of both flooding and wildfires (Agrawal et al., 2020). The data sources used in this study are presented in Supplementary Information Tables S1 and S2.

## 2.3 Baseline and future flood risk under different SSP-RCP scenarios

To evaluate the current flood risk state, a baseline scenario was set. This baseline scenario serves as a reference period against which changes in the state can be measured (Allwood et al., 2014). For the flood hazard component, runoff coefficient and rainfall data were obtained from 1971-2000, with the average value being used as the reference. Additionally, for the burnt area indicator, the relevant data pertained to the period from 2012 to 2020, accounting for the 8-year recovery period following wildfires. Regarding other indicators such as GDP and population density, the most recent available data was considered. Supplementary Information Tables S1 and S2 provide detailed data used in this research.

To assess future flood risk, the SSPs and RCPs were employed. The SSPs contain narratives describing how the future may unfold regarding demographics, economics, and the likelihood of achieving climate change mitigation and adaptation targets (Riahi et al., 2017). These pathways are useful for deriving data on population density, GDP, and institutional and eco-

nomic capacity, all of which are instrumental in evaluating flood risk. The RCPs, as precursors to the SSPs, primarily focus on calculating the concentration of greenhouse gases in the atmosphere. They look solely at greenhouse gas emissions and their radiative forcing effects (Riahi et al., 2017). Therefore, when the SSP and RCP scenarios are combined, they provide an estimate of future climate change and socio-economic conditions. This study utilized the SSP scenarios to define the socio-economic changes and the RCPs to estimate the climatic changes. We selected an optimistic outlook (SSP1-2.6) and a pessimistic one (SSP5-8.5). We combined the SSP scenarios with their corresponding RCP scenarios because not all indicators are provided for SSP scenarios. For instance, the FWI was only calculated based on the RCP scenarios (Di Giuseppe et al., 2018). Therefore, for SSP1-2.6 and SSP5-8.5 scenarios, the RCP2.6 data and RCP8.5 data were used, respectively. One should note that the radiative forcing associated with SSP1-2.6 and SSP5-8.5 is similar to that of RCP2.6 and RCP8.5 (O'Neill et al., 2017).

When projected data for some exposure and vulnerability indicators was unavailable, we made calculations and assumptions to bridge the gap. For example, the population density and the total GDP data were calculated using growth factors obtained from the International Institute for Applied Systems Analysis (IIASA), as outlined in Riahi et al. (2017) (Supplementary Information Table S2). These growth factors were applied to the latest available values derived from the baseline period (2020) to estimate future values. Different assumptions were made based on the SSPs for the distance from fire stations. For SSP1, which entails substantial increases in healthcare institutions and efforts to deal with natural hazards (Ebi, 2014; O'Neill et al., 2017), we anticipated that by 2100, we assumed an additional 10 fire stations will be established, bringing the total to 10 extra stations compared to baseline. In contrast, for SSP5, where investments in healthcare institutions are substantial but not as large as in SSP1, and where there are more frequent and intense climate change effects, particularly wildfires and floods due to increased fossil fuel investments, we foresee a somewhat different scenario. In 2100, we expected an additional seven fire stations to be established. For some indicators, such as soil type and topography, we assumed they remain constant and thus we utilized the most recent data.

## 2.4 Indicator risk classification

In this study, all flood risk indicators were reclassified into four risk classes i.e., low, moderate, high, and very high risk (Zhang et al., 2020). This data normalization step was necessary as the data was obtained from different sources and had inconsistent units which impeded their aggregation. For instance, the original burnt area map developed by the United States Department of Agriculture (USDA) has eight classes (Chen, 1994) (Supplementary Information Fig. S1a). In contrast, the runoff coefficient ($K$) is dimensionless, ranging from 0 to 1, with 0 indicating no runoff and 1 representing complete runoff (Supplementary Information Fig. S1b).

Table 1: Overview of risk classifications for the indicators

| Component | Indicator | Original unit | Class range | Risk level |
|-----------|-----------|---------------|-------------|------------|
|           |           |               | 0-50        | Low (1)    |
|           | Burnt area (WF) | hectares | Continued on next page | |

Hazard

| Component | Indicator | Unit | Class range | Risk level |
|---|---|---|---|---|
| | | | 50-121 | Moderate (2) |
| | | | 121-404 | High (3) |
| | | | $\geq 404$ | Very high (4) |
| | Fire Weather Index (FWI) | 0-100 | 0-11.2 | Low (1) |
| | | | 11.2-21.3 | Moderate (2) |
| | | | 21.3-38 | High (3) |
| | | | $\geq 38$ | Very high (4) |
| | Runoff coefficient ($K$) | 0-1 | $\leq 0.2$ | Low (1) |
| | | | 0.2-0.3 | Moderate (2) |
| | | | 0.3-0.4 | High (3) |
| | | | $\geq 0.4$ | Very high (4) |
| | Population density (PD) | Person/km$^2$ | $\leq 120$ | Low (1) |
| | | | 120-563 | Moderate (2) |
| | | | 563-4085 | High (3) |
| | | | $\geq 4085$ | Very high (4) |
| Exposure | Total weighted average GDP (EC) | Million EUR | $\leq 8.8$ | Low (1) |
| | | | 8.8-18.2 | Moderate (2) |
| | | | 18.2-33.5 | High (3) |
| | | | $\geq 33.5$ | Very high (4) |
| | Distance from highways (RO) | m | $> 2000$ | Low (1) |
| | | | 1500-2000 | Moderate (2) |
| | | | 500-1500 | High (3) |
| | | | $< 500$ | Very high (4) |
| | Distance from river (RV) | m | $> 1500$ | Low (1) |

| Component | Indicator | Unit | Class range | Risk level |
|---|---|---|---|---|
| | | | 500-1500 | Moderate (2) |
| | | | 250-500 | High (3) |
| | | | $< 250$ | Very high (4) |
| | Elevation (EL) | m | $\leq 958$ | Low (1) |
| | | | 958-1591 | Moderate (2) |
| | | | 1591-3384 | High (3) |
| | | | $\geq 3384$ | Very high (4) |
| | Slope steepness (SS) | degrees | $\leq 2$ | Low (1) |
| | | | 2-5 | Moderate (2) |
| | | | 5-9 | High (3) |
| | | | $\geq 9$ | Very high (4) |
| | Soil texture (ST) | Type | Sand | Low (1) |
| | | | Loam | Moderate (2) |
| | | | Sandy loam | High (3) |
| Vulnerability | | | Silty clay | Very high (4) |
| | Land use (LU) | Type | Forest | Low (1) |
| | | | Grassland | Moderate (2) |
| | | | Bare area and crop land | High (3) |
| | | | Urban and surface water | Very high (4) |
| | Saturated hydraulic conductivity (HC) | mm/day | $\geq 5289$ | Low (1) |
| | | | 3702-5289 | Moderate (2) |
| | | | 61-3702 | High (3) |
| | | | $\leq 61$ | Very high (4) |
| | GDP per capita (CA) | EUR | $\geq 28759$ | Low (1) |

| Component | Indicator | Unit | Class range | Risk level |
|---|---|---|---|---|
| | | | 24910-28759 | Moderate (2) |
| | | | 23083-24910 | High (3) |
| | | | $leq$ 23083 | Very high (4) |
| | Distance from fire stations (FS) | km | < 23 | Low (1) |
| | | | 23-42 | Moderate (2) |
| | | | 42-68 | High (3) |
| | | | > 68 | Very high (4) |

For the continuous variables, we used the Jenks Natural Breaks classification. The resulting reclassified maps are provided in Supplementary Information Figure S2 and Figure S3. For the categorical variables, we considered different assumptions. For instance, for the soil texture, we considered that the higher the infiltration rate, the lower the runoff will be, and vice versa (Berhanu et al., 2013). We modified 12 USDA soil texture classifications into four classes: Silty Clay, Sandy Loam, Loam, and Sand (Supplementary Information Fig. S3c). The normalization of land use and land cover was adapted from the classification of Cramer et al. (2020). However, it was assumed that croplands have high-risk values, instead of moderate when considering agricultural damages. Furthermore, it was assumed that the bare area has a high-risk class as bare soils are highly vulnerable for erosion and high runoff generation, leading to a high (flash) flooding potential (Mukherjee and Singh, 2020) (Supplementary Information Fig. S3d). Generally, land use and soil texture are integrated into the runoff coefficient. However, these indicators are considered in the flood risk assessment for the vulnerability component. To facilitate comparisons between future wildfire danger indicated by FWI and the burnt area, the original six risk classes used in the Copernicus dataset for FWI were reclassified into four classes consistent with other datasets used in this study (Berg et al., 2021). The detailed indicators' class range is described in Table 1.

## 2.5 Analytical hierarchy process

Weights were assigned to each indicator contributing to flood risk (Fig. 2), as these indicators contribute differently to flood risk. To accomplish this, we employed the Analytical Hierarchy Process (AHP), a widely utilized method in flood risk assessments (de Brito and Evers, 2016). AHP offers the advantage of structuring complex problems in a hierarchical and logical framework (Ha-Mim et al., 2022). The method was proposed by Saaty (1988) as a Multi Criteria Decision Making (MCDA) tool for decision-makers to make robust and flexible decisions by conducting pairwise comparisons between various indicators and ranking them based on their relative importance. Due to its flexibility, AHP is highly applicable in the GIS environment (Wu

et al., 2022). Within the AHP framework, we conducted detailed pairwise comparisons among the indicators involved in the three risk components. This was accomplished using a matrix, wherein scores reflecting relative importance were assigned.

Given that the weights for each indicator are unavailable, we interviewed seven experts in the field of natural hazards to assess the importance of different indicators contributing to flood risk within the AHP framework (Supplementary Information Table S5). These experts were selected based on their expertise on natural hazard risk (Experts 2, 3, 5), including flood (Experts 4, 6) and fires (Experts 1, 7), and their research background, such as technical (Experts 2, 3, 4) and non-technical fields (Experts 1, 5, 6, 7). The academic background varies among the experts, spanning from social, technical, and multidisciplinary expertise. Each expert has knowledge and experience related to wildfire risk, flood risk, or multi-risk assessments involving natural hazards. The experts' backgrounds were carefully considered to mitigate potential biases in scoring indicators, influenced by their fields of expertise (Zio, 1996). A template in the form of an Excel file obtained from Goepel (2013) was used to execute the AHP. The interview was guided providing detailed explanations during the AHP procedure. Furthermore, the consistency index and ratio were calculated to check the consistency of the experts' opinions and validate the weighting (Roy et al., 2021). Experts could re-evaluate their answers to ensure an appropriate consistency index in case of high inconsistencies. A detailed description of the AHP process can be found in the Supplementary Information (Supplementary Method).

After completing the AHP analysis, the normalized weights obtained were multiplied for each indicator and then aggregated to derive the flood index for each component. This process allows us to calculate the Flood Hazard Index (FHI), the Flood Exposure Index (FEI), and the Flood Vulnerability Index (FVI), as denoted by Equation 1. This approach is instrumental in flood risk assessment since not all indicators contribute equally to flood risk (Ghosh and Kar, 2018). Furthermore, to achieve a holistic assessment of flood risk, relative weights ($b1$ for flood hazard, $b2$ for exposure, and $b3$ for vulnerability) determined through the AHP analysis were assigned to the different components (hazard, exposure, and vulnerability) of flood risk (adapted from Zhang et al. (2020)). The total of these weights is 1 (see Supplementary Information and Section 3.4.1).

$$FR = b1 * FHI + b2 * FEI + b3 * FVI \tag{1}$$

Where FHI, FEI, and FVI are determined from hazard properties and socio-economic indicators as follows:

$$FHI = x1 * K + x2 * WF \tag{2}$$

$$FEI = y1 * EC + y2 * PD + y3 * RV + y4 * RO \tag{3}$$

$$FVI = z1 * EL + z2 * SS + z3 * ST + z4 * LU + z5 * HC + z6 * CA + z7 * FS \tag{4}$$

The constants $xi$, $yi$, and $zi$ ($i$ being the number of the indicator) are the weights for each flood indicators obtained from the AHP. K and WF in the FHI are runoff and wildfires denoted by the FWI or burnt area, respectively. The FEI consists of

EC, PD, RV, and RO, representing GDP per region, population density, distance from the river, and distance from highways, respectively. Vulnerability (FVI) has seven indicators: EL for elevation, SS for slope steepness, ST for soil texture, LU for land use, HC for hydraulic conductivity, CA for economic capacity, and FS for distance from the fire station.

All the weighting values for flood risk, hazard, exposure, and vulnerability components obtained from the AHP could be applied to other river basins, including smaller ones. This suggests that additional expert interviews are unnecessary if the proposed approach is applied to other areas as long as all the flood components remain the same. However, additional expert interviews become imperative if additional exposure and vulnerability indicators are introduced. Furthermore, the inclusion of experts with different backgrounds may lead to slightly different weighting values, although we anticipate the difference will be relatively small, as has been shown by other studies investigating the sensitivity of weightings (de Brito et al., 2019; Moreira et al., 2023).

## 3   Results

Flood risk (FR) is defined as a function of flood probability, exposure, and vulnerability, which are represented in the flood hazard index (FHI), flood exposure index (FEI) and flood vulnerability index (FVI), respectively (Eq. 1) (Klijn et al., 2015). Thus, FR can be estimated prior to calculating FHI, FEI, and FVI, which are described in Section 3.1 for FHI, Section 3.2 for FEI, and Section 3.3 for FVI. The FR is described in Section 3.4.

### 3.1   Distribution of the Flood Hazard Index (FHI)

The AHP analysis results in different weights for runoff and wildfires. The burnt area (WF) receives 30% of the weight and runoff (K) receives 70% of the weight for the FHI calculation, considering the wildfire effect (Eq. 2). In the scenario where the effect of wildfires on flood risk was not taken into account, the runoff receives a full 100% weight in the FHI calculation. Interestingly, all experts acknowledged that wildfires tend to increase the runoff due to reduced soil infiltration capacity (Supplementary Table S5). Most participants mentioned that runoff was crucial for flood hazard assessment, primarily because wildfires generally occur in localized areas within the catchment. However, the runoff occurs in the entire catchment. Thus, the experts strongly favor a higher weight for the runoff. They also argued that burnt areas do not lead to flood risk without a water source. Only one expert argued that the two indicators are equally important since these indicators are strongly interlinked, where each could exacerbate the other. Another respondent strongly emphasized the wildfire factor over the runoff, citing personal experiences with flash flooding in burnt areas as a compelling reason. The weight percentage of hazard prioritization is presented in Supplementary Figure S4.

Figure 3a illustrates the FHI map, considering the effects of wildfires. The majority of the FHI values fall within the low-risk category. Notably, the influence of burnt areas is visible, leading to increased FHI values in areas where wildfires have occurred (Supplementary Information Fig. S1a). Major cities in the region mostly have low FHI values, with only Pamplona falling into the moderate-risk category. Supplementary Information Figure S5 clearly shows the increase of future FHI under various scenarios, both with and without wildfire effects. The effect of wildfires on FHI is more pronounced for the SSP5-

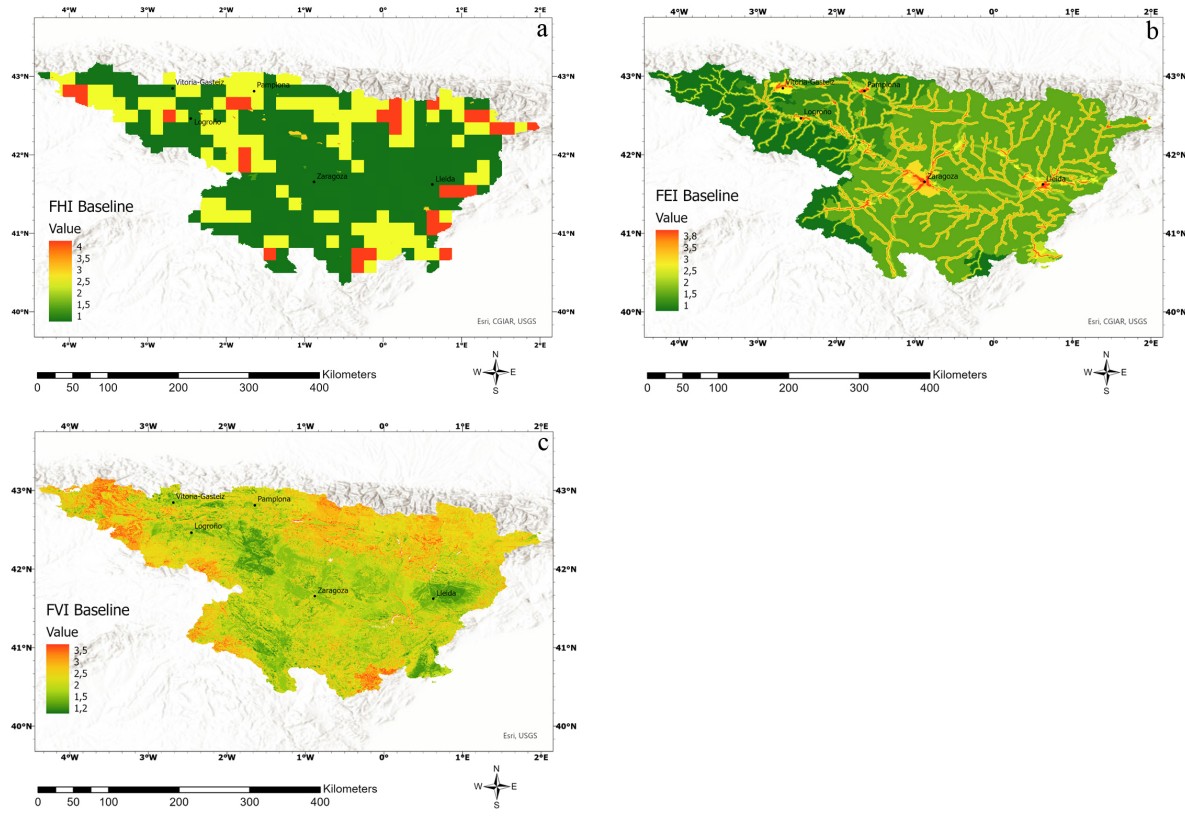

**Figure 3.** a) Spatial distribution of the classification of the FHI for the baseline scenario, b) spatial distribution of the classification of the FEI for the baseline scenario, and c) spatial distribution of the classification of the FVI for the baseline scenario. Green color (FHI=1) indicates low risk, light green color (FHI=2) indicates moderate risk, orange color (FHI=3) indicates high risk, and red color (FHI=4) indicates very high risk. Regions with white contour are outside the studied region.

8.5 scenario, even though the runoff coefficient declines in the RCP8.5 scenario (Supplementary Information Fig. S6b). This discrepancy is attributed to the strong increase in the FWI in RCP8.5, which counteracts the decline in runoff when considering wildfire effects. The increase in FHI without accounting for wildfire effects in the RCP8.5 scenario is explained as the runoff receives full weight (100%) in this scenario while in the baseline scenario with wildfire effect, runoff is counted only 70%.

### 3.2 Distribution of the Flood Exposure Index (FEI)

The distribution of weight for the FEI, derived from expert judgments, shows that population density (PD) and the distance from the river (RV) both receive the highest portion of the weight (37%). The GDP (EC) receives the second-largest share of weight (18%), while the distance from highways (RO) accounts for only 8% of the total weight (Eq. 5) (Supplementary Information Figure S7). These weight distributions were used to calculate the FEI, as illustrated in Figure 3b. The main reason

for the notably higher weight for the population density and distance from the river is that saving lives is the most important factor of the exposure component and in the flood risk assessment (Supplementary Information Table S5). One respondent emphasized the general hierarchy in operational risk management, where prioritizing people's safety is of utmost importance over infrastructure and nature. The high weight given to distance from rivers (37%) reflects that proximity to rivers poses a significant risk to human lives and economic assets. One expert noted that in mountainous areas within the basin, low population density is attributed to the local population's awareness of flash flood-prone zones, leading them to avoid exposure. Conversely, exposure to highways was not considered a critical indicator by all respondents, largely because this factor is already incorporated into the GDP, which is viewed as a more comprehensive indicator by the experts.

$$FEI = 0.18 * EC + 0.37 * PD + 0.37 * RV + 0.08 * RO \qquad (5)$$

Figure 3b clearly indicates the high exposure levels in major cities due to the high population density. Moreover, these cities are centered along river streams, which make them susceptible to flood risk. On the other hand, the western part of the basin exhibits lower exposure levels, which are attributed to its lower total GDP indicator, as shown in Supplementary Information Figure S2b. The maximum value of the FEI for the baseline is 3.8, whereas future scenarios yield maximum values of 4. This increase in the FEI results from projected population and economic growth, depicted in Supplementary Information Figure S8, elevates the overall exposure levels. Notably, the SSP5-8.5 scenario substantially increases the FEI due to its strong population and total GDP growth.

### 3.3 Distribution of the Flood Vulnerability Index (FVI)

The FVI is determined based on weight distributions for each vulnerability indicator and in our case is highly diverse. Among the indicators, Slope Steepness (SS) receives the highest weight, accounting for 26% (Supplementary Information Figure S9). The economic capacity (CA) shown in GDP per capita follows closely with 22% of the weight while land cover/land use (LU) represents 17% of the weight. The Distance from Fire Stations (FS), Saturated Hydraulic Conductivity (HC), Soil Texture (ST), and Elevation (EL) receive 11%, 9%, 8%, and 8% of the weight, respectively (Eq. 6). Interview results highlight different opinions among experts regarding the importance of vulnerability indicators (Supplementary Information Table S5). However, most participants agreed that slope steepness holds the utmost significance due to its strong correlation with increased runoff and (flash) flood risk on steep slopes. Slope steepness also impacts wildfire risk, as wildfires can spread more easily on such terrain, according to some wildfire experts. Figure 3c shows regions with steep slopes (Supplementary Information Fig. S3b) having high FVI values. Additionally, most experts consider GDP per capita crucial because vulnerability can be considerably reduced if there are high economic resources to adapt to wildfires and flooding. Our finding on the importance of slope steepness and GDP is in agreement with previous studies. Roy et al. (2021) found that elevation and slope are the most important indicators for flood risk while Moreira et al. (2021) also found that social and economic indicators, such as population and income are crucial in flood vulnerability. Here, we classified the population as exposure, which also receives the highest weight. The distance from fire stations indicator is deemed less important because most experts believe that fire stations

do not possess the capacity to fully address flooding. Most experts also acknowledge the land use indicator to be important due to its significant influence on runoff and wildfires.

$$FVI = 0.08 * EL + 0.26 * SS + 0.08 * ST + 0.17 * LU + 0.09 * HC + 0.22 * CA + 0.11 * FS \qquad (6)$$

The last three geophysical indicators related to vulnerability, namely hydraulic conductivity, soil texture, and elevation, are generally perceived as of low importance by most experts. An explanation would be that some experts may lack sufficient knowledge about the soil texture and hydraulic conductivity indicators, while others may prioritize social aspects of vulnerability and the slope steepness indicator in the FVI. Moreover, many experts note a significant correlation between hydraulic conductivity and soil texture, which could lead to redundancy in their inclusion. Opinions on the elevation indicator are rather diverse. Some experts believe that water accumulation in low-lying areas results in high vulnerability, while others argue that flash flooding can also occur in higher areas. Some experts consider SS a more comprehensive indicator for assessing vulnerability, which could influence their views on the inferior importance of elevation in the FVI.

Figure 3c provides a clear visualization of the importance of each indicator in developing the FVI map. The significance of slope steepness, as shown in Supplementary Information Figure S3b, is clearly observable, with areas around the Pyrenees and Iberian mountains showing high FVI values. On the other hand, the delta of the basin, the region around the city of Lleida, and the area between Zaragoza and Logroño predominantly exhibit low FVI values. This is attributed to the absence of slope steepness, coupled with high economic and institutional capacity reflected in GDP per capita and the presence of fire stations, as shown in Supplementary Information Figure S3b, S3g, and S3h, respectively.

For the future scenarios with and without the wildfire effect, changes in the FVI values are similar as shown in Supplementary Information Figure S10. The primary factor that distinguishes FVI values in scenarios with and without wildfire effects is hydraulic conductivity. Although hydraulic conductivity values are lower with wildfire effects in the future scenarios compared to baseline (Supplementary Information Fig. S11 and Fig. S3e,f), this variable only counts a 9% weight in the total FVI calculation (Eq. 6). This explains the overall similarity between FVI values with and without the wildfire effect. However, the maximum value of the FVI increases from 3.5 to 3.6 for scenarios without wildfire effects, which can be explained by the changes in land use and saturated hydraulic conductivity. Furthermore, the wildfire effects increase the maximum FVI value to 3.6 in 2100 for SSP5-8.5 (Supplementary Information Fig. S10d). This is primarily due to a significant increase in FWI for this scenario.

### 3.4 Flood Risk map (FR)

#### 3.4.1 Weight distribution for FR map

After all flood risk (FR) component maps, such as FHI, FEI, and FVI were developed, we analyzed the FR map based on the weighting distribution for each FR component ($b1$, $b2$, and $b3$ in Eq. 1) obtained from the AHP analysis. The FVI receives 46% of the weight, FEI takes up to 34% of the total weight distribution, and the FHI only receives 20% of the total weight

**Table 2.** Reclassification of spatial distribution of flood risk for the FR equation with corresponding color indication

| Risk level | Class value | Flood probability class | Color indication |
|:---:|:---:|:---:|:---:|
| Low | 1-1.5 | ≤16.7% | green |
| Moderate | 1.5-2 | 16.7-33.3% | light green |
| High | 2-2.5 | 33.3-50% | orange |
| Very high | 2.5-4 | ≥50% | red |

distribution (Eq. 7)(Supplementary Information Figure S12). The FVI receives almost half the weight from the experts because they believe vulnerability is the most manageable risk component. Another reason is that societies experience negative effects from flooding and wildfires when they are vulnerable. Here, we agreed with the experts' opinions that flood risk could be better managed if the vulnerability is reduced. This can be achieved through increasing societal interventions, such as improving economic and institutional capacities, and physical interventions such as slope stabilization, regreening, and soil improvement works. Exposure is also considered crucial since risk cannot exist without exposure and both people and economic assets, which are part of FEI, play essential roles in flood risk assessment. However, exposure and hazard are less manageable compared to vulnerability.

$$FR = 0.2 * FHI + 0.34 * FEI + 0.46 * FVI \qquad (7)$$

The experts' insights highlight the different perspectives on flood and wildfire risk management (Supplementary Information Table S5). Fire experts emphasized that fire hazard is more manageable, primarily due to effective fuel management practices, while flood management is viewed as rather complex. Interestingly, many technical experts underscored the importance of FEI and FVI over FHI because they believe understanding the societal consequences of flooding is key to effective risk management. They expressed the need for a holistic approach that considers all risk components, including exposure and vulnerability, rather than just focusing on the hazard. However, one expert who has a background in social science and works in some technical aspects advocated for equal weight distribution (around 33%) to all risk components, emphasizing that risk assessment and management are multifaceted problems with both technical and social aspects.

In this study, the FR map was categorized into four distinct classes, each representing a different flood risk level. These categories were defined based on the flood risk values, with a score of less than 1, meaning low risk, and a score of 4, meaning very high risk (Table 2). Since the flood risk is assessed for the annual mean, the class range values are unevenly distributed, with a larger class range for the 'Very high' risk level. Any flood risk level of 50% or higher during the year was categorized as 'very high' flood risk. The other three flood risk values were divided equally into intervals of 16.7% per class.

### 3.4.2 Flood risk for the baseline scenario

Figure 4a shows the FR map for the baseline scenario developed using Equation 7. The map clearly highlights the relevance of indicators, such as population density, slope steepness, and distance from the river in assessing flood risk. High and very

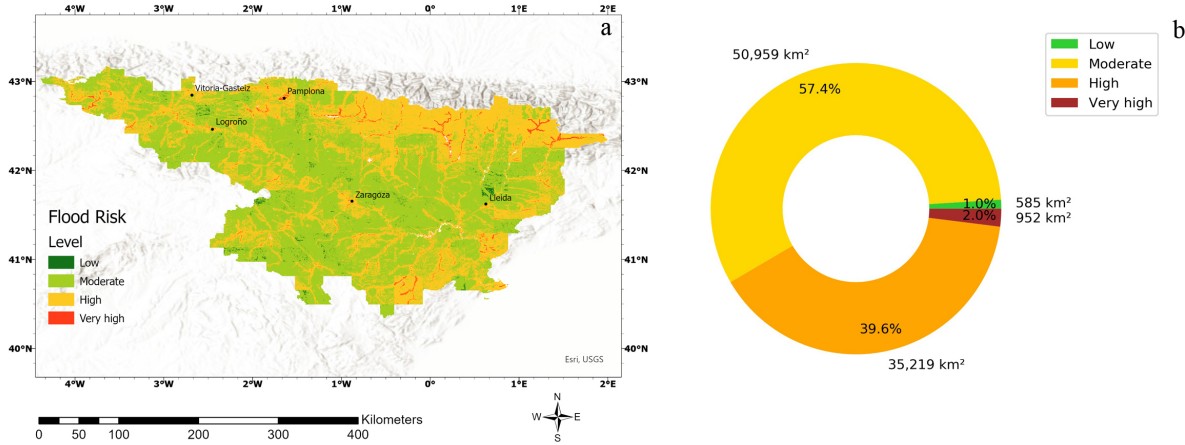

**Figure 4.** a) Flood risk map for the baseline scenario for the Ebro River basin and b) distribution of flood risk classes corresponding to surface area in km$^2$

high flood risk areas are prominently identified, particularly in the mountainous regions in the northern part of the basin, the city of Pamplona, and in the delta of the basin. These areas are recognized as hotspots for (flash) flood risk. Additionally, the northeastern part of the basin exhibits high flood risk, which can be attributed to factors such as a high runoff coefficient, exposure in terms of total GDP, steep slopes, low saturated hydraulic conductivity, and limited institutional capacity, as depicted in Supplementary Information Figure S1b, S2b, S3b, S3e, and S3g, respectively. On the other hand, some areas have low flood risk values in various regions, owing to the complex interplay of indicators that makes it challenging to explain. Overall, only 1% of the area, covering 585 km$^2$, falls into the low flood risk category, while another 1%, equivalent to 952 km$^2$ falls into the very high flood risk category (Fig. 4b). The majority of the basin exhibits moderate and high flood risk levels, covering 58% (50,959 km$^2$) and 40% (35,219 km$^2$) of the total basin, respectively. Notably, the effects of burnt area, which were evident in the FHI for the baseline (Supplementary Information Fig. S1a), are less pronounced in the overall FR map. This might be attributed to the lower weight assigned to the burnt area indicator (30%) in the FHI. Moreover, the FHI component is only counted for 20% in the FR calculation, limiting the influence of burnt area on the total flood risk assessment.

### 3.4.3 Flood risk for future scenarios without wildfire effects

Compared to the baseline scenario, the flood risk map for the year 2100 under SSP5-8.5 reveals hotspots of high and very high flood risk, particularly in the southern and eastern parts of the basin (Fig. 5a). These high and very high flood risks in these regions result from a strong increase in exposure, including GDP, population density, and vulnerability as indicated by the land use indicator. The increase in flood risk is less apparent for SSP1-2.6, except in areas like Pamplona and the southern region of Leida (Supplementary Information Fig. S13a). Overall, the proportion of the basin characterized by moderate flood risk increases from 58% to 65% (57,011 km2) for SSP1-2.6 (Fig. 4b and Supplementary Information Fig. S13b). On the contrary,

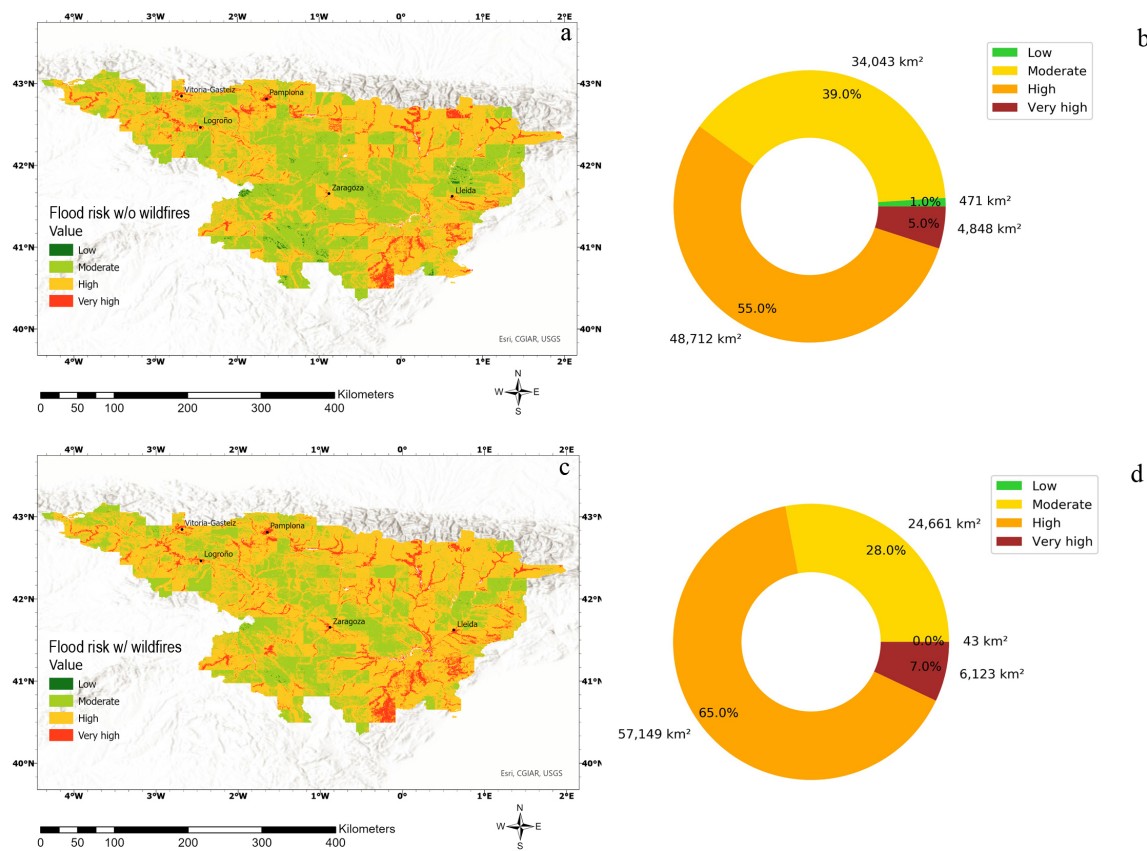

**Figure 5.** a) Flood risk map without wildfire effects in the Ebro River basin for SSP5-8.5 year 2100, b) distribution of flood risk classes without wildfire effect for SSP5-8.5 year 2100 corresponding to surface area in km$^2$, c) same as a but with wildfire effect, and d) same as b but with wildfire effect.

the area that may experience high flood risk in the future decreases from 40% for the baseline scenario to 32% for SSP1-2.6 due to a reduction in FVI (Supplementary Information Fig. S10a). For the SSP5-8.5 scenario, more than half (55%) of the basin falls into high flood risk category (48,712 km$^2$), with 5% (4,848 km$^2$) classified as very high flood risk, indicating a substantial increase in high flood risk in 2100 (Fig. 5b). The main driver behind the increase in flood risk for SSP5-8.5 is the increase of all three flood risk components (FHI, FEI, and FVI) (Supplementary Information Fig. S5b, S8b, and S10b, respectively).

### 3.4.4 Flood risk for future scenarios with wildfire effects

Figure 5c clearly shows the effect of wildfires on future flood risk. Wildfires lead to a 2% increase in the probability of very high flood risk and a 10% increase in the probability of high flood risk for SSP5-8.5 (Fig. 5b and 5d). This increase in flood risk when wildfires are considered in the analysis is caused by the increase in FHI due to the use of FWI (Supplementary Information Fig. S5d). As previously described, the strong increase in FWI in the RCP8.5 scenario, which reduces saturated

hydraulic conductivity, explains the increase in FHI. Conversely, the change in FVI is negligible (Supplementary Information Fig. S10b,d). The hotspot regions classified as very high flood risk remain relatively consistent with and without wildfires. Compared to the baseline scenario, the total areas with very high and high flood risk nearly doubled, rising from 41% to 72% (Fig. 4b and 5d).

The influence of wildfires on flood risk is less pronounced for SSP1-2.6 (Supplementary Information Fig. S13c). Wildfires reduce the areas classified as having moderate flood risk from 65% to 60% and shift these regions towards high-risk flood areas from 32% to 38% (Supplementary Information Fig. S13b,d). Interestingly, the area that will be categorized as prone to very high and high flood risk under SSP1-2.6 (39%) is slightly smaller than in the baseline scenario (41%). This can be attributed to the slightly lower FVI under SSP1-2.6 compared to the baseline (Supplementary Information Fig. S10c).

## 4 Discussion

### 4.1 Effect of wildfires on floods based on baseline scenario

Analyzing the effect of wildfires on flood clearly indicates the amplifying effect, with higher risk when wildfires are considered in the analysis as agreed by many experts (Fig. 3). Wildfires may alter the hydraulic conductivity and thus increase runoff (Seibert et al., 2010; Folador et al., 2021). In an Italian basin, the runoff response due to wildfires increased from 75% to 125% after wildfires occurred. A similar finding is also found in an American basin, with a 120% increase in runoff response compared to pre-fire runoff value (Seibert et al., 2010). During the interviews, experts also emphasize that wildfires lead to an increase in runoff coefficient. In some cases, wildfires showed no impact on runoff response but in most cases, they increased runoff response by a factor of 1.2 to 6.5 during heavy rainfall events (Leopardi and Scorzini, 2015). The observed effect of wildfire in increasing runoff could be used to replace the expert interview. For example, an increase in the runoff coefficient by a factor of 1.2 after wildfires would increase the FHI by 1.2 (FHI=1.2 x runoff) and equal to one for scenario without wildfires (FHI = 1 x runoff). This approach, however, requires using a rainfall runoff model, measurement, or experiment to obtain the correction factor. In this research, a strong increase in flood hazard in burnt areas is not particularly evident. This may be related to the limited occurrence of large-scale wildfires in the Ebro Basin for the baseline scenario (see Supplementary Information Fig. S1a). The Mediterranean ecosystem typically has low biomass, resulting in a decreased fuel load and fuel moisture, contributing to the limited occurrence of large wildfires (Bedia et al., 2013; Meyn et al., 2007).

### 4.2 Future flood risk by taking into account wildfires

Analyzing future flood risk is always subject to high uncertainties due to the estimation of future flood risk components, such as flood probability, exposure, and vulnerability. In the case of the FHI, the main components are runoff and wildfires. The projection of runoff, which is crucial for analyzing the FHI, strongly depends on precipitation patterns. In the Mediterranean region, it is expected that although there will be more heavy rainfall events, there will also be a decrease in the number of precipitation days, resulting in lower average runoff, especially under the RCP8.5 scenario (Erol and Randhir, 2012; Shakesby,

2011). When analyzing future runoff, it is important to consider both precipitation and human activities, such as agricultural and industrial water demand, flood protection measures like dams and weirs, and land use and land cover management (García-Ruiz et al., 2011). However, these human activities are typically considered in the exposure and vulnerability components of flooding rather than in the hazard component.

In this study, we also acknowledge that analyzing future FHI while considering the effects of wildfires introduces additional uncertainty. There is no definitive data to predict future fires, so the study relies on the FWI as a proxy for the probability of fire occurrences, which has been shown to have a substantial effect on the FHI, especially for the SSP5-8.5 scenario (Bedia et al., 2013). Some research indicates that the number of dry days is expected to increase significantly in the Mediterranean, potentially leading to higher wildfire risk in the future (Erol and Randhir, 2012; Hoinka et al., 2007; Ruffault et al., 2020). However,

predicting future wildfire risk is intrinsically more complex and wicked. The majority of wildfires are largely human-provoked, compared to floods which are induced by human activity, its impact depends, for example, on forest (fuel) management, vegetation and land use practices, and fuel moisture (Shakesby, 2011; Turco et al., 2014). Wildfire risk varies spatially and can vary depending on the climate change scenarios. For example, SSP1-2.6 assumes better fuel and land use management, leading to a lower wildfire hazard than SSP5-8.5 (Wu et al., 2015). However, some argue that forest management in the Mediterranean is

largely unmanaged and requires improved management to deal with wildfires and, indirectly, flooding (Lindner et al., 2010). Yet, the majority of wildfires in the region are human-ignited, which shifts the focus of forest management towards risk communication to societies to prevent human-caused wildfires (Martínez et al., 2009). Naturally, adverse climatic conditions can exacerbate the effects of human-induced fires by increasing the size, severity, and frequency of devastating wildfires in the future (Pausas and Keeley, 2021).

Socio-economic changes also play an important role in this research, particularly in assessing future wildfires and flood risk by considering exposure and vulnerability components. The IIASA database for climate change was used, yielding a higher exposure for SSP5-8.5, as population and economic growth are higher compared to SSP1-2.6 (Riahi et al., 2017). In terms of vulnerability, previous studies have shown severe impacts of increased vulnerability across the physical aspects of the Mediterranean, such as land-use changes in terms of desertification and urbanization, pose huge threats to the Mediterranean

societies (Erol and Randhir, 2012; Filipe et al., 2013; Cramer et al., 2018). Furthermore, with regard to flood vulnerability, the limited capacity of stormwater management systems to cope with changes in flash flooding patterns in the future, along with an increase in populated flood-prone regions, are expected to result in a higher vulnerability in most parts of the Mediterranean (Cramer et al., 2018). Rapid urbanization makes communities more vulnerable to the impacts of flooding caused by climate change, often due to a lack of awareness and the ineffectiveness of policies and management in communicating and mitigating

risk (Llasat, 2021). Other drivers, including deforestation and socio-economic inequalities, further exacerbate the susceptibility of societies to flooding (Papathoma-Köhle et al., 2021). However, the increase in vulnerability may not be clearly evident in the FVI analysis, possibly due to the effective management of wildfires and flooding, which is considered with the inclusion of distance to fire stations as one of the indicators. Additionally, some indicators are assumed to remain relatively constant in the future. Furthermore, land-use changes for SSP1-2.6 in the far future may have an opposite relationship with flooding due to a

significant expansion of forested areas (García-Ruiz et al., 2011). Hence, it is crucial to consider the adaptive capacity of the

population to deal with the adverse effects of climate change regarding wildfires and flooding, which are shown to be relatively high in the Ebro River basin, with minor spatial differences in economic and institutional capacity within the basin.

## 4.3 Shifting from a single hazard paradigm to multi-hazard approach

The effect of wildfires on flood risk, particularly concerning climate change, underlines the urgency of effectively managing these interconnected risks. Currently, hazards like wildfires and floods are often managed separately rather than using a multi-risk approach. This siloed approach can lead to ineffective and inefficient risk management, especially when these hazards can occur successively (de Ruiter et al., 2020). Despite their potential interdependence, the experts who participated in this research also highlighted the challenges of managing wildfires and flood risks as separate entities. Some experts stated that sometimes institutions coping with wildfires and floods have different perspectives and management plans and see these hazards as individual events. This underscores the need for interdisciplinary collaboration to create comprehensive multi-hazard risk management strategies to address the increasing threat of wildfires and their alteration effects on flood risk.

A multi-hazard approach, which involves conducting multi-risk assessments, has been widely advocated for and is considered fundamental in building robust governance and management structures to address the increase of compound and cascading natural hazards (Arosio et al., 2020; de Ruiter et al., 2020). This study focuses on an integrated approach, considering the multi hazards of wildfires and flooding, which aligns with this recommendation. It is important to note that wildfires can have various impacts, such as erosion leading to water quality issues and landslides triggered by heavy rainfall events. Therefore, there is a need to assess and understand the effects of wildfires more comprehensively (Pouyan et al., 2021). Therefore, this research provides an example of how to integrate multiple hazards into risk evaluation by conducting comprehensive assessments that consider numerous drivers and indicators that will contribute to increased flood risk in the future.

## 4.4 Data limitations

Assessing future flood risks is a complex task that relies on historical and projection data related to flood hazard, exposure, and vulnerability. These data are not always available; therefore, some assumptions need to be made. We acknowledge that the amplifying effect of wildfires on flood risk is considered only for on-the-spot impact. The analysis did not incorporate the downstream effects of increased runoff on bare land due to wildfires. This could potentially underestimate the true extent of the effect of wildfires on flood risk. The flood risk assessment was conducted annually, even though wildfire and flood occurrence can vary substantially within seasons. We also did not consider the flood return periods since the runoff coefficient was used. As mentioned in Section 4.2, using FWI may overestimate the occurrence of wildfires. The FWI was estimated solely based on the meteorological variables, influencing fire activity or fuel dryness. Thus, the FWI only captures the potential fuel moisture and does not explicitly model fire evolution (Di Giuseppe et al., 2018; Abatzoglou et al., 2019). One should note that FWI is a measure of fire danger and fire ignition is required to start a fire (Van Wagner, 1987). This study did not consider the distinction between human-induced and natural fires. Fires are complex hazards mostly triggered by human ignitions rather than solely depending on climate conditions (Versini et al., 2013; Sutanto et al., 2019). Therefore, this study focused on the natural variables that set wildfire risk and enable a spark to build into a wildfire without considering if it was human or natural

induced. Furthermore, the recovery time of wildfires was assumed to be 8 years for the baseline and is not considered in the future projection. The actual recovery time for burned areas can vary widely depending on factors like vegetation type and post-fire management practices. We also acknowledge that the number of experts involved in this study is limited. AHP relies on the experts' judgements and therefore can be subjective. Different experts might provide evaluations based on their personal experiences and biases. This is also found in our result where one expert gave more weight to wildfires than runoff due to their experience in flash floods on burnt areas.

## 5 Conclusions

This study employed an integrated multi-criteria GIS-based approach to assess flood risk in the Ebro River basin in Northern Spain, considering current and future scenarios, wildfire effects, and socio-economic drivers. Interviews with experts highlighted the consensus that vulnerability and exposure are the most critical components in a flood risk assessment. However, experts have various reasons for prioritizing certain indicators and components contributing to flood risk. Their backgrounds and experiences in natural hazards and disasters explain various reasons. In the baseline scenario, the study found that the effects of burnt areas did not significantly contribute to an increase in flood risk, due to the limited occurrence of large wildfire events during the study period. Instead, indicators such as the runoff coefficient, population density, land use and land cover, and slope steepness played a significant role in flood risk, particularly in major cities and the Pyrenees region.

Compared to the baseline scenario, SSP1-2.6 for the year 2100 shows a reduction in flood risk, even when not considering the effects of wildfires. This reduction can be attributed to significant developments in adaptive capacity, including increased economic and institutional resources, leading to improved resilience to flooding. Furthermore, lower exposure levels and less severe climate change impacts in this scenario contribute to lower flood risk. The strongest increase in flood risk is apparent for SSP5-8.5 for the year 2100, primarily due to substantial population growth, urbanization, and lower institutional resources to cope with flooding. It is evident that a strong increase in flood risk is intensified when considering the wildfires, with a significant increase in wildfire risk contributing to the high flood risk. This study highlights the importance of adopting a multi-hazard risk management approach, as solely focusing on individual risks may underestimate multiple hazards' compound and cascading effects. The integrated flood risk assessment conducted here provides valuable insights into the complex dynamics of flood risk, emphasizing the need for comprehensive strategies to build resilience against the increasing frequency of extreme weather events and their associated risks.

*Data availability.* Data used in this study are freely available online from many sources. We provided information on the data source in the Supplementary Table A1 and A2.

*Author contributions.* S.J.S., M.J., and M.P.G. conceived and implemented the research. Data analyses and all figures have been performed by M.J. M.M.B. contributed to interpreting the results and discussion. All authors contributed substantially to the editing and commenting on the article drafts for several rounds. All authors have read and agreed to the published version of the manuscript.

*Competing interests.* The authors declare no conflict of interest.

*Acknowledgements.* This research is supported by the ML-CDHEU project, which is funded by the WUR Data Driven Discoveries in a Changing Climate investment (D3C2) project code 5160958747. The authors would like to thank all experts for their contribution as without them the current study would not have been possible. This research supports the work of the IAHS Helping program Drought in the Anthropocene (DitA).

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
