# Peer review of "The effect of wildfires on flood risk: A multi-hazard flood risk approach for Ebro River basin Spain"

_EGUsphere, 2024_

## Author Comment (AC1)

**Reply to reviewer 1**

We would like to thank the reviewer for careful reading throughout our manuscript and for valuable suggestions and comments. In this document, we reply to each of these. **L** refers to the line number. For example, **L65-70**, refers to lines 65-70.

| Reviewer 1 | | |
|---|---|---|
| **No** | **Comment** | **Reply** |
| 1 | This study presents an approach for considering related risks for wildfires and floods which is applied to produce baseline and future predictions for a single case study (Ebro river basin). This risk assessment method utilises the Fire Weather Index (FWI) and a number of indicators which were weighted according to expert feedback via an Analytical Hierarchy Process approach. The importance of considering the cascading, interlinked risks of fires and floods are clearly outlined and the chosen case study provides a useful demonstration and context. | We would like to thank reviewer for the acknowledgement of our manuscript that it provides a valuable demonstration and context of cascading floods and fires. |
| 2 | However, further information about the expert panel decision-making (in addition to the information provided in the supplementary info) would be helpful given the important role it plays in the final risk assessment method and the chosen weightings. In particular, I would welcome more information regarding the variability in opinions offered by the experts (ideally quantified to show the variability around the final, chosen weightings at each stage). | The referee has a valid point regarding more detailed AHP results. We will include a summary of the AHP results on exposure, hazard, and vulnerability prioritization in the revised Supplementary Information.

Furthermore, we have incorporated expert opinions concerning the selected weights in the main text, specifically in the result section e.g., opinions about runoff and fire weights in **P11L225-P12L231**. In the revised version, we will provide further elaboration and reference these statements to the AHP results. |
| 3 | Additionally, it may be useful to provide further context in the results/discussion by explicitly comparing the expert panel opinions to the existing literature where possible. | We thank the feedback from the reviewer. However, it is important to note that detailed weights for all indicators used in this study are not available from the references. Hence, we conducted the AHP analysis (**P11L204-205**). In the literature, only a few articles were found discussing the increase of runoff due to wildfire in percentages (e.g., Folador et al., 2021; Leopardi and Scorzini, 2015) (**P18L356-362**). However, they did not discuss the weighting factors for analyzing flood risk. Our study is pioneering in flood risk analysis by considering the cascading effect of wildfires. We will add a summary on the expert opinions in the revised manuscript to provide further clarification. |
| 4 | Additionally, a large part of the fire activity is characterized by FWI predictions and it is important that the meaning of these predictions, the historical context of this index and it's role as a fire danger prediction tool, along with limitations if | We appreciate the reviewer suggestions regarding the FWI predictions and their limitations. This was discussed in **P18L377-P19L391**. In the revised manuscript, the discussion about FWI will be expanded and |

| | | |
|---|---|---|
| | looking to extrapolate expected fire occurrence from FWI are clearly outlined. This is addressed in some of the references cited in the manuscript as highlighted in the specific comments below. | additional references will be added accordingly. |
| 5 | Line 21: Is there a more appropriate reference here than Wilby and Keenan which so far as I can tell does not address the link between fire and drought? | We appreciate the reviewer for spotting the mismatch. We will address this by adding references Mazdiyasni and AghaKouchak (2015) and He et al. (2022) in the revised version, which discuss the increase of drought, dry spell, and heatwaves. |
| 6 | Lines 30-31: Double check this statistic in the provided reference. Is this based on the info given at the start of the introduction in this reference? If so this is actually only over the last 20 years which may be worth highlighting. Also the % of the population affected seems a bit higher for flooding with droughts having affected 25% of the population. | In this sentence, we described the global population affected by floods, which was estimated around 2.5 billion people (2.5/8.1*100=30%) over the last 20 years, according to Tabari et al. (2021). We will revise the sentence. |
| 7 | Lines 44-46: Perhaps rephrase this section since one of the conclusions of Versini et al 2013 study is that 'our assumptions can appear as a low hypothesis that should underestimate the impact of forest fire on the hydrological response'. This seems to contradict the claim here that this study in an exception in not underestimating the amplification effects. Perhaps this is more about being understudied or receiving little consideration in which case this could be clarified in-text. | The referee has a valid point regarding the impact of wildfires in flood risk assessment. We will rewrite the sentence into: "Concerning the third source of complexity, despite amplifying the risk of floods, the impacts induced by wildfire are often given little consideration in conventional flood risk assessments". |
| 8 | Lines 73-74: Could clarify that Balasch et al. state that this was the mean figure for the period of 1920-2000. | Indeed, the mean precipitation of 622 mm is averaged from the period of 1920-2000. We will add this information in the revised manuscript. |
| 9 | Lines 82-83: Can you clarify/ rephrase for clarity here? I think from what's written in Terrado et al, that 38 people/km^2 is the average population density of the basin, rather than the average density for these 2 largest cities. | We agreed with reviewer that the population density of the whole basin is 38 people/km$^2$. We will clarify this statement in the revised version. |
| 10 | Lines 91-93: There may be specific motivations for dealing with wildfire management at various spatial scales. Is there other evidence that can also be provided here to support the statement that 'flood management appears to have a higher priority than fire management' e.g. a comparison of spending/funding? | At least in Europe, we have flood directive on the assessment and management of flood risks (2007/60/EC of the European Parliament). However, we could not find any information regarding fires directive although there are policies to protect the EU's forests against fire. We will add this information in the revised version. |
| 11 | Line 100: Are these previous literature reviews published and available to cite here? | In this sentence, when referring to the literature review, we meant the process of data collection to obtain all the indicators employed in this study. The sources of these data are presented in Supplementary Table S1 and S2. We will revise the text accordingly. |
| 12 | Lines 104-105: Can you clarify that Fire Weather Index provides a prediction of fire | We thank the reviewer for the suggestions. We will revise the statement from "future fire |

| | | |
|---|---|---|
| | danger? And perhaps accompanied by clarification of the distinction from 'probability' which will also be affected by other factors e.g. the limitations highlighted in Abatzoglou et al, 2019.

In this context, is 'probability of future fire events' a suitable term? As the probability (including likelihood of ignition) will vary with other factors e.g. location relative to population centers, public access, social dimensions which are not considered in a meteorological index which predicts the fire weather and danger were a fire to occur.

See also the discussion in Di Giuseppe et al 2018 which you have cited. e.g. pg 5360 'The FWI is already widely employed in fire management and control (Lee et al., 2002). However, it does not explicitly model fire evolution, but it is a measure of fire danger (Van Wagner, 1987). Even for extreme FWI values there is a need for a stochastic component, i.e. ignition, to start a fire. For this reason, situations in which FWI is high but no fire is recorded are not uncommon' | events" to "prediction of fire danger". Additionally, the paragraph will be expanded to incorporate a detailed description of FWI.

Regarding the suggestion about the likelihood of ignition, we have discussed this in **P18L378-P19L391.** We will include the suggested limitation factors of FWI in the revised manuscript.

Furthermore, we will incorporate the limitations of FWI as described in Di Giuseppe et al. (2018) into the new version of the paper. |
| 13 | Line 116: As discussed later, were only highways considered? If so can you clarify this here when introducing the distance from roads parameter. | In this study, we only considered the distance from the highways as one of the exposure components since they play a major role in transportation (**P6L123-124**). We will describe this information here. |
| 14 | Lines 117-120: Is there any way to further assess the validity of this weighting approach? e.g. by further exploring the heterogeneity of economic activity in some of these regions e.g. by using population as a proxy for this or incorporating a distance from major town/city element? | Indeed, the reviewer has a point here. However, assessing the total GDP per province based on factors such as the types of economic activities in the province, distance from major towns/cities, or population can complicate the analysis. A more straight forward approach would be to calculate the weighting of GDP for each province based on the area in $m^2$ that lies inside the basin. |
| 15 | Lines 122-123: Was there a particular reason for the choice of this cut-off length or is this choice arbitrary? | We used 50 km as a proxy for the length to exclude small streams and only obtain the main channels that contribute more to flood vulnerability. It is arbitrary. |
| 16 | Lines 123-124: In relation to an earlier comment, were these the only roads considered? If so can this be clarified earlier when distance from roads is first mentioned (line 116). Perhaps this parameter could even just be labelled 'distance from highways' throughout or 'distance to major roads' as in Roy et al 2021. | We agreed with the reviewer and thus, we will clearly indicate distance from highways, following Roy et al. (2021). We will also modify the label from distance from road to distance from highways. |
| 17 | Lines 141-143: Are the size/number of personnel at fire stations considered at all? Is number of fire stations a better indicator (and/or easier to analyse) than for example total spending on fire resources? | In this study, we focus solely on the number of fire stations as the institutional capacity indicator, and hence we do not consider the number of personnel and budget. These indicators are not freely available online and |

| | | |
|---|---|---|
| | Indeed, McLennan and Birch outline some of the complexity involved in the prevention and management stages alone in their discussion of various factors including station staff size, average age of firefighting staff, degree of co-operation between staff, additional private firefighting resources.

Could you provide some further discussion of the suitability and limitations of number of fire stations as an indicator? | difficult in quantification for future scenarios. The reviewer also acknowledges this complexity, as described in McLennan and Birch (2005). For our baseline scenario, we use the most recent number of fire stations available. However, for future scenario, we need to make assumptions on the number of future fire stations. For SSP1, we assume 10 fire stations will be established while only 7 stations will be established for SSP5 (**P7L168-174**). |
| 18 | Lines 165-175: Can you clarify if all assumptions are listed here? If not could you provide a full list of unavailable exposure/vulnerability indicators and corresponding assumptions made e.g. in the Supplementary data.

Can you explain how these assumptions were chosen? Does this involve the previously mentioned expert interviews or is this an arbitrary choice? | We provide all indicator assumptions in Supplementary Table S2, including the links to obtain the datasets. For many indicators, we calculate the future scenarios using growth factors derived from IIASA SSP public database version 2 (**P7L166-167**). Thus, we do not arbitrarily choose the number of future indicators, rather, we base our projections on established growth factors. |
| 19 | Lines 191-192: Would danger be a better word here than effects? Or perhaps just refer only to fire weather index? | We thank the reviewer for spotting typo. Here we meant future wildfire effect, indicated by FWI. |
| | Line 205: Understand the potential need for anonymity around experts involved but is it possible to provide any further details about specific areas/extent of expertise? | We will indicate their fields of expertise in the revised version. |
| 20 | Lines 227-231: How do these expert comments relate to any existing approaches in the literature?

Given these complex considerations, is the scenario well-defined enough for experts to pass judgment on relative importance? In the future, would more local scale analysis involving expert analysis be required? Or for example, could a further set of scenarios be designed for expert feedback in which additional factors could be incorporated e.g. different landscape types, distance from river. Understanding of course that these are considered in other parts of the model. | In general, all experts agreed that wildfires could increase runoff, and this was confirmed by previous studies (Seibert et al., 2010; Folador et al., 2021) (**P18L357-358**). However, to the authors' knowledge, no study has discussed the weighting of runoff and wildfires for analyzing the flood hazard index. This was the primary reason for conducting the AHP.

The scenario for weighting all the indicators are well-defined and we guided the whole AHP processes. We also calculated the consistency index to check whether answers of experts are consistent and the consistency ratio to make sure the respondents' weighting is consistent and validated (Roy et al., 2021) (Supplementary Method). For future research including local scale analysis, more exposure and vulnerability indicators could be included in the analysis for baseline study. For future scenarios, however, one should take into consideration on the available datasets or other reasonable assumptions need to be made. We will elaborate this information in the discussion section. |
| 21 | Line 232: For comparison, could you also show burnt area and runoff maps in this | We will combine Supplementary Figure S1 with Figure 3 in the revised manuscript. |

| | | |
|---|---|---|
| | figure? So that the influence of the chosen weighting can be understood. | |
| 22 | Lines 237-239: So is it an increase in burnt area as a result of increased FWI which results in the increased FHI? | Yes, a strong increase in the FWI in RCP8.5 leads to higher wildfire component and thus FHI. We do not state burnt area since it is a prediction of wildfires and not a real burnt area, unlike in the baseline scenario. |
| 23 | Lines 242-244: Could more information about the variability in weighting assigned by the expert panel be provided? | We will provide the results of AHP in the Supplementary Information. |
| 24 | Lines 266-267: Was this the area in which the greatest difference in expert opinion was observed? | The highest difference in expert opinion is found in exposure indicators, such as population density (7.8%) and distance from river (6%). The figures of weight percentages including their error bars will be provided in Supplementary Information. |
| 25 | Lines 272-273: How does this compare to findings in the existing literature as were outlined in the introduction to this study? | Although distance from fire station is deemed less important than slope steepness and economic capacity, it is ranked as the third highest above saturated hydraulic conductivity, soil texture, and elevation (**P13L265-266**). |
| 26 | Lines 273-274: Would greater consensus have been reached by asking experts only to consider a smaller number of more relevant indicators? | We believe if the number of indicators is smaller, the expert will still weight the indicators based on the importance and thus, we will obtain the same rank but with different weights. |
| 27 | Lines 283-286: Are there existing studies regarding the influence of hydraulic conductivity which can provide further context for the decisions of the expert panel? | Yes. Versini et al. (2013) show an increase in river discharge after forest fire occurred in the Llobregat river basin, Spain (**P6L127-128**). Moreover, Seibert et al. (2010) and Folador et al. (2021) also show an increase in runoff after wildfire occurred (**P18L357-358**). |
| 28 | Lines 296-297: Can you explain how decision fits within the context of understanding and mapping existing risk (prior to any other management interventions) and for predictions based upon assigned levels of societal intervention. | We agree with the experts that FVI should get more weight and then followed by the exposure and hazard. The flood risk could be better managed if the vulnerability is reduced by increasing societal interventions such as improving economic and institutional capacities and physical interventions such as slope, regreening, and soil improvement works. Hazard and exposure, on the other hand, are less manageable. |
| 29 | Lines 301-303: What was the break-down of expertise in the panel? How significant was this difference in perspective? | We will provide the expertise of the experts in the revised manuscript. The background of the experts shows a different perception on the flood risk. As it was explained, fire experts consider fuel management practices as a manageable component while flood experts consider wildfires are a wicked problem due to human influence (ignition). We will discuss this accordingly. |
| 30 | Lines 306-307: What was the break-down of expertise in the panel? How significant was this difference in perspective? | The expert who indicates that exposure and vulnerability should be equally weighted has a background in social science and works on some technical aspects. This shows that experts' perspective is influenced by their backgrounds and their working environment. |

| | | We will incorporate this in the revised version. |
|---|---|---|
| 31 | Lines 324-325: Can you clarify how notable this finding is? Or whether in fact this is just entirely due to the weightings assigned by the expert panel? | The burnt area is weighted for 30% in the flood hazard index (**P11L222**). Moreover, flood hazard index is only counted 20% in the calculation of flood risk compared to flood exposure index and flood vulnerability index (Eq. 4, **P15L300**). Thus, the effect of wildfires is greatly reduced in the flood risk calculation. We will clarify this in the revised version. |
| 32 | Lines 342-343: Given the role of FHI, how much is the effect of wildfires controlled by the choice of weighting for burnt area: runoff? | For future flood risk analysis, the most influential factor is the FWI and not the weights assigned for wildfire (30%) and runoff (70%). Strong increase in FWI for RCP8.5 counteracts the decrease in runoff. |
| 33 | Lines 356-357: How much does it indicate this vs. indicating the perceived role cascading effect of wildfires given the role of the expert panel in determining the various weightings? | All the experts agreed that burnt area affects runoff and therefore most of them gave high weight to runoff. |
| 34 | Lines 360-362: How possible would it be in future studies to further incorporate these previous findings to augment or replace the need for expert weightings? | A simple method to replace expert weightings is by using a correction factor. For example, if the runoff increases by a factor of 1.2 after wildfire occurred as found in Leopardi and Scorzini (2015), then the FHI is equal to 1.2 x runoff and FHI is equal to 1 x runoff for evaluating FHI without wildfire. We will discuss this in the revised version. |
| 35 | Lines 364-366: As discussed in Bedia et al 2013, does fuel moisture (and/or linked meteorological conditions) also play a role in limiting these large fire events? | Yes, the occurrence of wildfires largely depends on forest (fuel) management, vegetation and land use practices, and fuel moisture (**P19L382-384**). |
| 36 | Lines 378-379: As per previous comments, can you explicitly address the limitations of using FWI as a proxy for fire probability? | We discussed this limitation in **P20L434-438**. |
| 37 | Line 432: How did the annual timeframe affect the chosen FWI? Was this an average value for a whole year? Or a maximum value? | The FWI employed in the study is the mean fire weather index value over the European fire season (seasonal FWI, June-September) for 2050 and 2100 averaged from multi models. See Supplementary Table S2 for detailed information. |
| 38 | Section 4.4: Can you also discuss any limitations involved with the expert panel and the Analytical Hierarchy Process and associated data? | The limitation of AHP and expert panel will be discussed in the revised version. |
| 39 | Technical Corrections
Figure 2: Proofing comment - check figure quality/resolution as slightly blurred in places.
Line 149: Typesetting issue '1971 -2000'
Line 179: 'USDA' - Acronym needs to be defined on 1st use.
Line 312: Typo: 'into intervals of 16,7% per class'
Line 365: Typo: 'can be related to that the Mediterranean' | We thank for the feedback on Figure 2. We will check the quality of the figure.

The space will be added.

We will add the acronym of USDA.

We will change comma with dot.

We will rephrase the sentence accordingly. |

**Additional References:**

Folador, L., Cislaghi, A., Vacchiano, G., and Masseroni, D.: Integrating Remote and In-Situ Data to Assess the Hydrological Response of a Post-Fire Watershed, Hydrology, 8(4), 169, https://doi.org/10.3390/hydrology8040169, 2021.

Leopardi, M. and Scorzini, A.: Effects of wildfires on peak discharges in watersheds [Technical Reports], iForest - Biogeosciences and Forestry, 8(3), 302–307, https://doi.org/10.3832ifor1120-007, 2015.

Mazdiyasni, O., and AghaKouchak, A.: Substantial increase in concurrent droughts and heatwaves in the United States, PNAS, 112(37), 11484-11489, https://doi.org/10.1073/pnas.1422945112, 2015.

He, B., Zhong, Z., Chen, D., Liu, J., Chen, Y., Miao, C., Ding, R., yuan, W., Guo, L., Huang, L., Hao, X., and Chen, A.: Lengthening dry spells intensify summer heatwaves, Geophysical Research Letters, 49, https://doi.org/10.1073/pnas.1422945112, 2022.

Tabari, H., Hosseinzadehtalaei, P., Thiery, W., and Willems, P.: Amplified Drought and Flood Risk Under Future Socioeconomic and Climatic Change, Earth's Future, 9(10), e2021EF002 295, https://doi.org/10.1029/2021EF002295, 2021.

Di Giuseppe, F., Rémy, S., Pappenberger, F., and Wetterhall, F.: Using the Fire Weather Index (FWI) to improve the estimation of fire emissions from fire radiative power (FRP) observations, Atmos. Chem. Phys., 18(8), 5359–5370, https://doi.org/10.5194/acp-18-5359-2018, 2018.

Roy, S., Bose, A., and Chowdhury, I. R.: Flood risk assessment using geospatial data and multi-criteria decision approach: a study from historically active flood-prone region of Himalayan foothill, India, Arabian Journal of Geosciences, 14(11), 999, https://doi.org/10.1007/s12517-021-07324-8, 2021.

McLennan, J. and Birch, A.: A potential crisis in wildfire emergency response capability? Australia's volunteer firefighters, Global Environ- mental Change Part B: Environmental Hazards, 6(2), 101–107, https://doi.org/10.1016/j.hazards.2005.10.003, 2005.

Seibert, J., McDonnell, J. J., and Woodsmith, R. D.: Effects of wildfire on catchment runoff response: a modelling approach to detect changes in snow-dominated forested catchments, Hydrology Research, 41(5), 378–390, https://doi.org/10.2166/nh.2010.036, 2010.

Versini, P. A., Velasco, M., Cabello, A., and Sempere-Torres, D.: Hydrological impact of forest fires and climate change in a Mediterranean basin, Natural Hazards, 66(2), 609–628, https://doi.org/10.1007/s11069-012-0503-z, 2013.

Leopardi, M. and Scorzini, A.: Effects of wildfires on peak discharges in watersheds [Technical Reports], iForest - Biogeosciences and Forestry, 8(3), 302–307, https://doi.org/10.3832ifor1120-007, 2015.

---

## Author Comment (AC2)

**Reply to reviewer 2**

We would like to thank the reviewer for valuable suggestions and comments. In this document, we reply to each of these. **L** refers to the line number. For example, **L65-70**, refers to lines 65-70.

| Reviewer 2 | | |
|---|---|---|
| 1 | This study shows an assessment of flood risk using a multi-criteria GIS-based approach incorporating wildfires and floods for the Ebro basin. The study tackles an important topic; it is a generally well written and well-argued paper, with a strong narrative and clear structure. | We would like to thank reviewer for the acknowledgement of the importance of the topic and the clarity of the paper. |
| 2 | At points in the paper, some terms are used somewhat interchangeably and not always defined. For example, in the title the authors use "effects", then later "impacts", then a mix. Similarly, what is meant by a cascade in this paper? Is it a cascade in terms of a trigger or something that increases risk (i.e. a mechanism or process that links these two hazards even if temporally), or a cascade in terms of impacts, or both? I feel as those terms are being used somewhat interchangeably in this paper.

For example, in L12 in the abstract, the authors say "…especially when considering the cascading impacts of wildfires". I would question here what the cascade is? Or indeed, whether this is an impact? To me, this perhaps is more of knock on effect or something that increased the risk of something else through changing vulnerabilities (as the authors note later on) – a wildfire affects the flood risk through burnt area and so forth, which in turn may cause impacts for example – but is this process an impact? I would suggest that clearly defining these terms and then staying with them throughout would benefit the understanding for the reader. | We thank the reviewer for raising the terminology used in the paper. We agree with the reviewer and decide to use the term effect instead of impact. Moreover, we will provide the definition of cascading used in the manuscript, which is triggering or amplifying the flood risk. We will change this term throughout the revised manuscript. |
| 3 | The abstract mentions indicator only once, but a large part of the study is actually focused towards the integration of socio-economic indicators and land-use change information with 'conventional' hydrological properties and the cascading effects of wildfire to assess flood risk. This is a complicated endeavor within a multi-hazard/multi-risk approach, which is good to see, but I think the fact that this is approach should be made much clearer in the abstract and title so it is clear that the story is not solely about the cascading effects of wildfires and flood, it is more about better flood risk | The referee has a valid point regarding mentioning indicators. In the revised version, we will elaborate more in the indicators used in the study. Moreover, we will also expand the abstract highlighting the multi-hazard/risk approach and how this could be done for better flood risk assessment.

We thank the reviewer for the suggestion on the title and for stressing out the multi-hazard/risk instead of cascading. We will revise the title and abstract accordingly. |

| | | |
|---|---|---|
| | assessment as a whole, incorporating wildfires.

Later, in section 4.3 in the discussion, the authors state "Therefore, this research provides an example of how to integrate multiple hazards into risk evaluation by conducting comprehensive assessments that consider numerous drivers and indicators that will contribute to increased flood risk in the future." – this is a, I believe, a better (more correct?) framing for the study. I would suggest multi-hazard/risk be included in some way in the title, perhaps even removing the word cascade, and a reduction on the focus of the wildfire and flood cascades and more towards flood risk using multiple inputs from the start. | |
| 4 | The methods are comprehensive but section 2.2 is framed around flood risk indicators, however here this is where wildfire risk and indicators (FWI for example) is employed. Related to my point above, this section to me should be given a clear multi-hazard focus towards flood risk to make it clear that fires are part of this. Some subtle reframing and – importantly – including wildfire in the subtitle may be beneficial to guide the reader. | We thank the reviewer for the suggestion. We will reframe the context of the article toward the focus of multi-hazard/risk. We will modify the heading of chapter 2.2 into "Indicators for flood risk assessment including wildfire risk". |
| 5 | The use and placement of the equations based on expert judgement FR, FEI etc is very confusing. FR is at the end of the methods section but if not defined until section 3.4. Then, additional equations, such as FVI, appear later on. Lines 213-4 says "This process allows us to calculate the Flood Hazard Index (FHI), the Flood Exposure Index (FEI), and the Flood Vulnerability Index (FVI), as denoted by Equation 1", however equation 1 shows the equation for FR. Equations 2 and 3 are not referenced from the text. Some terms, such as FS, are really hard to find the definitions of (one has to go looking in the text), and no units are provided. The use and presentation of these needs a rethink.

I would suggest that the equations are all placed in the methods and defined there, leaving the results to focus on the weighting by expert judgment and, therefore, the outcomes of the study. Indeed, many of the sub-sections 3.3 and 3.4 for example, stray into methods rather than results. Some careful reordering would really help the readability and accessibility. | The reviewer expresses concern about the use of equations based on the expert judgement and the structure of the results. Indeed, we place the results of FR in chapter 3.4 because FR cannot be calculated without knowing the FHI, FEI, and FVI values beforehand. This is the reason that we describe the results of FHI (chapter 3.1), FEI (3.2), and FVI (3.3) first. In the method, we present the main formula of calculating the FR, which consists of FHI, FEI, and FVI components. We will provide all the flood risk component formulas in the chapter 2.5 when we revise the paper, including their full names (not abbreviation). In addition, we will provide a short paragraph in between chapters 3 and 3.1 explaining the structure of the results. |
| 6 | Related to my above point, there is a large emphasis on expert judgement of the | We thank the reviewer for the feedback. We will provide the results of AHP in the |

| | | indicator weightings. It is not clear though quite how much emphasis they have on the results. Is seven people enough? Are they all from the Erbo region? Does this matter? The earlier phases of the study are quite analytical, but then the focus moves to a judgement based approach. Additional details on this process, perhaps in section 2.5, would in turn help understand and interpret the later results section. | Supplementary Information. Moreover, the background of the experts will be provided in the chapter 2.5 in the revised version. Indeed, we only managed to interview seven experts although we sent interview requests more than seven. Some experts declined with various reasons and even one of them does not agree with the multi-risk framework that we formulated for the study. We will discuss the limitation of only interviewing seven experts in the revised manuscript. |
|---|---|---|---|
| 7 | The discussion is good and very readable. It provides some excellent additional information. I do think though that perhaps a bit more work may be needed to separate the location-specific findings based on local expert judgement and the wider findings that can be employed elsewhere. The authors don't really attempt to do this; instead the assumption is that the results shown here for Erbo would hold elsewhere. This localised expert judgement is not mentioned in the limitations in 4.4. The title at the start calls this paper a "study case" but really Erbo is the study here primarily. Making it clear that the findings for the Erbo maybe separate from broader interpretations would be beneficial in the discussion, including some detail on how this can be done (and the limitations of doing so), would really elevate this to be a usable example more broadly. | We thank the reviewer for the suggestion. Indeed, our study is only for the Ebro River basin. However, the approach employed in this study could be applied elsewhere as long as the data are available. The experts are not only coming from Spain but also other countries in Europe. The expert background will be expanded in the revised manuscript. We will make it clear about the applicability of our approach. |
| 8 | The conclusions in section 5 state "The research underscores the need for interdisciplinary collaboration…" in relation to the experts. While I agree with this statement, this is not a focus of the study and isn't presented up to this point. If the authors which to explore this narrative, this would be better placed in the discussion above. | The reviewer has a valid point and we will move the sentence into discussion (chapter 4.3). |
| 9 | Specific/minor comments:

Line 191-2    Please clarify or reword what is meant by future wildfire effects (FWI) and the burnt area? FWI is already defined, but what is meant by effects? | We thank the reviewer for spotting typo. Here we meant future wildfire effect, indicated by FWI. |

---

## Author Response (AR1)

**Reply to reviewer 1**

We would like to thank the reviewer for careful reading throughout our manuscript and for valuable suggestions and comments. In this document, we reply to each of these. **L** refers to the line number. For example, **L65-70**, refers to lines 65-70.

| Reviewer 1 | | |
|---|---|---|
| **No** | **Comment** | **Reply** |
| 1 | This study presents an approach for considering related risks for wildfires and floods which is applied to produce baseline and future predictions for a single case study (Ebro river basin). This risk assessment method utilises the Fire Weather Index (FWI) and a number of indicators which were weighted according to expert feedback via an Analytical Hierarchy Process approach. The importance of considering the cascading, interlinked risks of fires and floods are clearly outlined and the chosen case study provides a useful demonstration and context. | We would like to thank reviewer for the acknowledgement of our manuscript that it provides a valuable demonstration and context of cascading floods and fires. |
| 2 | However, further information about the expert panel decision-making (in addition to the information provided in the supplementary info) would be helpful given the important role it plays in the final risk assessment method and the chosen weightings. In particular, I would welcome more information regarding the variability in opinions offered by the experts (ideally quantified to show the variability around the final, chosen weightings at each stage). | The referee has a valid point regarding more detailed AHP results. We included a summary of the AHP results on exposure, hazard, vulnerability, and risk components prioritizations in the revised Supplementary Information (**Figure S4, S7, S9, S12**). Furthermore, we have incorporated expert opinions concerning the selected weights in the main text, specifically in the result section e.g., opinions about runoff and fire weights in **P11L225-P12L231** (previous version). In the revised version, we provided further elaboration and experts' statements in the **Supplementary Table S5** (see example in **P13L268**). |
| 3 | Additionally, it may be useful to provide further context in the results/discussion by explicitly comparing the expert panel opinions to the existing literature where possible. | We thank the feedback from the reviewer. However, it is important to note that detailed weights for all indicators used in this study are not available from the references. Hence, we conducted the AHP analysis (**P11L222-224**). In the literature, only a few articles were found discussing the increase of runoff due to wildfire in percentages (e.g., Folador et al., 2021; Leopardi and Scorzini, 2015) (**P20L417-421**). However, they did not discuss the weighting factors for analyzing flood risk. Our study is pioneering in flood risk analysis by considering the cascading effect of wildfires. We added a summary on the expert opinions in the revised manuscript to provide further clarification (**Supplementary Table S5**). |
| 4 | Additionally, a large part of the fire activity is characterized by FWI predictions and it is important that the meaning of these | We appreciate the reviewer suggestions regarding the FWI predictions and their limitations. In the revised manuscript, the |

| | | |
|---|---|---|
| | predictions, the historical context of this index and it's role as a fire danger prediction tool, along with limitations if looking to extrapolate expected fire occurrence from FWI are clearly outlined. This is addressed in some of the references cited in the manuscript as highlighted in the specific comments below. | explanation about FWI was expanded and additional references was added accordingly (**P5L112-117**). |
| 5 | Line 21: Is there a more appropriate reference here than Wilby and Keenan which so far as I can tell does not address the link between fire and drought? | We appreciate the reviewer for spotting the mismatch. We addressed this by adding references Mazdiyasni and AghaKouchak (2015) and He et al. (2022) in the revised version, which discuss the increase of drought, dry spell, and heatwaves (**P1L22**). |
| 6 | Lines 30-31: Double check this statistic in the provided reference. Is this based on the info given at the start of the introduction in this reference? If so this is actually only over the last 20 years which may be worth highlighting. Also the % of the population affected seems a bit higher for flooding with droughts having affected 25% of the population. | In this sentence, we described the global population affected by floods, which was estimated around 2.5 billion people (2.5/8.1*100=30%) over the last 20 years, according to Tabari et al. (2021). We revised the sentence (**P2L31-32**). |
| 7 | Lines 44-46: Perhaps rephrase this section since one of the conclusions of Versini et al 2013 study is that 'our assumptions can appear as a low hypothesis that should underestimate the impact of forest fire on the hydrological response'. This seems to contradict the claim here that this study in an exception in not underestimating the amplification effects. Perhaps this is more about being understudied or receiving little consideration in which case this could be clarified in-text. | The referee has a valid point regarding the impact of wildfires in flood risk assessment. We rewrote the sentence into: "Concerning the third source of complexity, despite amplifying the risk of floods, the impacts induced by wildfire are often given little consideration in conventional flood risk assessments" (**P2L45-47**). |
| 8 | Lines 73-74: Could clarify that Balasch et al. state that this was the mean figure for the period of 1920-2000. | Indeed, the mean precipitation of 622 mm is averaged from the period of 1920-2000. We added this information in the revised manuscript (**P3L76**). |
| 9 | Lines 82-83: Can you clarify/ rephrase for clarity here? I think from what's written in Terrado et al, that 38 people/km^2 is the average population density of the basin, rather than the average density for these 2 largest cities. | We agreed with reviewer that the population density of the whole basin is 38 people/km$^2$. We clarified this statement in the revised version (**P3L84-85**). |
| 10 | Lines 91-93: There may be specific motivations for dealing with wildfire management at various spatial scales. Is there other evidence that can also be provided here to support the statement that 'flood management appears to have a higher priority than fire management' e.g. a comparison of spending/funding? | At least in Europe, we have flood directive on the assessment and management of flood risks (2007/60/EC of the European Parliament). However, we could not find any information regarding fires directive although there are policies to protect the EU's forests against fire. We added this information in the revised version (**P4L94-99**). |
| 11 | Line 100: Are these previous literature reviews published and available to cite here? | In this sentence, when referring to the literature review, we meant the process of data collection to obtain all the indicators employed in this study. The sources of these data are presented in Supplementary Table |

| | | S1 and S2. We revised the text accordingly (**P5L106-108**). |
|---|---|---|
| 12 | Lines 104-105: Can you clarify that Fire Weather Index provides a prediction of fire danger? And perhaps accompanied by clarification of the distinction from 'probability' which will also be affected by other factors e.g. the limitations highlighted in Abatzoglou et al, 2019.

In this context, is 'probability of future fire events' a suitable term? As the probability (including likelihood of ignition) will vary with other factors e.g. location relative to population centers, public access, social dimensions which are not considered in a meteorological index which predicts the fire weather and danger were a fire to occur.

See also the discussion in Di Giuseppe et al 2018 which you have cited. e.g. pg 5360 'The FWI is already widely employed in fire management and control (Lee et al., 2002). However, it does not explicitly model fire evolution, but it is a measure of fire danger (Van Wagner, 1987). Even for extreme FWI values there is a need for a stochastic component, i.e. ignition, to start a fire. For this reason, situations in which FWI is high but no fire is recorded are not uncommon' | We thank the reviewer for the suggestions. We revised the statement from "future fire events" to "probability of fire danger prediction" (**P5L112**).

Regarding the suggestion about the likelihood of ignition, we have discussed this in **P20L445-P21L448.** We included the suggested limitation factors of FWI in the revised manuscript. Furthermore, we also incorporate the limitations of FWI as described in Di Giuseppe et al. (2018) into the new version of the paper (**P22L500-507**). |
| 13 | Line 116: As discussed later, were only highways considered? If so can you clarify this here when introducing the distance from roads parameter. | In this study, we only considered the distance from the highways as one of the exposure components since they play a major role in transportation. In this study, we did not consider provincial and local roads as one of exposure indicators. We described this information here (**P5L136-138**). |
| 14 | Lines 117-120: Is there any way to further assess the validity of this weighting approach? e.g. by further exploring the heterogeneity of economic activity in some of these regions e.g. by using population as a proxy for this or incorporating a distance from major town/city element? | Indeed, the reviewer has a point here. However, assessing the total GDP per province based on factors such as the types of economic activities in the province, distance from major towns/cities, or population can complicate the analysis. A more straight forward approach would be to calculate the weighting of GDP for each province based on the area in $m^2$ that lies inside the basin (**P5L130-134**). |
| 15 | Lines 122-123: Was there a particular reason for the choice of this cut-off length or is this choice arbitrary? | We used 50 km as a proxy for the length to exclude small streams and only obtain the main channels that contribute more to flood vulnerability. It is arbitrary. We added the word arbitrarily for better clarification (**P5L136**). |
| 16 | Lines 123-124: In relation to an earlier comment, were these the only roads considered? If so can this be clarified earlier when distance from roads is first mentioned (line 116). Perhaps this parameter could even just be labelled | We agreed with the reviewer, and we clearly indicated distance from highways, following Roy et al. (2021). We also modified the label from distance from road to distance from highways e.g., figure 2, Table 1 and revised the text accordingly. |

| | | |
|---|---|---|
| | 'distance from highways' throughout or 'distance to major roads' as in Roy et al 2021. | |
| 17 | Lines 141-143: Are the size/number of personnel at fire stations considered at all? Is number of fire stations a better indicator (and/or easier to analyse) than for example total spending on fire resources?

Indeed, McLennan and Birch outline some of the complexity involved in the prevention and management stages alone in their discussion of various factors including station staff size, average age of firefighting staff, degree of co-operation between staff, additional private firefighting resources.

Could you provide some further discussion of the suitability and limitations of number of fire stations as an indicator? | In this study, we focus solely on the number of fire stations as the institutional capacity indicator, and hence we do not consider the number of personnel and budget. These indicators are not freely available online and difficult in quantification for future scenarios (**P7L157-158**). The reviewer also acknowledges this complexity, as described in McLennan and Birch (2005). For our baseline scenario, we use the most recent number of fire stations available. However, for future scenario, we need to make assumptions on the number of future fire stations. For SSP1, we assume 10 fire stations will be established while only 7 stations will be established for SSP5 (**P8L187-191**). |
| 18 | Lines 165-175: Can you clarify if all assumptions are listed here? If not could you provide a full list of unavailable exposure/vulnerability indicators and corresponding assumptions made e.g. in the Supplementary data.

Can you explain how these assumptions were chosen? Does this involve the previously mentioned expert interviews or is this an arbitrary choice? | We provide all indicator assumptions in Supplementary Table S2, including the links to obtain the datasets (**P8L183-184**). For many indicators, we calculate the future scenarios using growth factors derived from IIASA SSP public database version 2 (**P8L182-185**). Thus, we do not arbitrarily choose the number of future indicators, rather, we base our projections on established growth factors. |
| 19 | Lines 191-192: Would danger be a better word here than effects? Or perhaps just refer only to fire weather index? | We thank the reviewer for spotting typo. Here we meant future wildfire danger indicated by FWI (**P8L210**). |
| | Line 205: Understand the potential need for anonymity around experts involved but is it possible to provide any further details about specific areas/extent of expertise? | We indicated the experts' fields of expertise in the revised version (**P11L224-226**). |
| 20 | Lines 227-231: How do these expert comments relate to any existing approaches in the literature?

Given these complex considerations, is the scenario well-defined enough for experts to pass judgment on relative importance? In the future, would more local scale analysis involving expert analysis be required? Or for example, could a further set of scenarios be designed for expert feedback in which additional factors could be incorporated e.g. different landscape types, distance from river. Understanding of course that these are considered in other parts of the model. | In general, all experts agreed that wildfires could increase runoff, and this was confirmed by previous studies (Seibert et al., 2010; Folador et al., 2021) (**P20L418-419**) (**Supplementary Table S5**). However, to the authors' knowledge, no study has discussed the weighting of runoff and wildfires for analyzing the flood hazard index. This was the primary reason for conducting the AHP with experts.

The scenario for weighting all the indicators are well-defined and we guided the whole AHP processes. We also calculated the consistency index to check whether answers of experts are consistent and the consistency ratio to make sure the respondents' weighting is consistent and validated (Roy et al., 2021) (**P11L230-P12L233**). For future research, local scale analysis involving expert analysis might be and not be needed. If more |

| | | or less exposure and vulnerability indicators are included in the analysis, then the interview should be performed again. Moreover, adding more experts with different backgrounds also may lead to different weighting values although we believe the difference will be relatively small. We elaborated this information in the AHP section (**P12L251-257**). |
|---|---|---|
| 21 | Line 232: For comparison, could you also show burnt area and runoff maps in this figure? So that the influence of the chosen weighting can be understood. | We would like to keep Supplementary Figure S1 in the appendix because this figure is not the main finding. On the hand, we refer Supplementary Figure S1 in the main text when we discuss figure 3 (**P13L277**). |
| 22 | Lines 237-239: So is it an increase in burnt area as a result of increased FWI which results in the increased FHI? | Yes, a strong increase in the FWI in RCP8.5 leads to higher wildfire component and thus FHI. We do not state burnt area since it is a prediction of wildfires and not a real burnt area, unlike in the baseline scenario. |
| 23 | Lines 242-244: Could more information about the variability in weighting assigned by the expert panel be provided? | We provided the results of AHP in the Supplementary **Figure S4, S7, S9, S12**. |
| 24 | Lines 266-267: Was this the area in which the greatest difference in expert opinion was observed? | The highest difference in expert opinion is found in exposure indicators, such as population density (7.8%) and distance from river (6%). The figures of weight percentages including their error bars were provided in Supplementary **Figure S4, S7, S9, S12**. |
| 25 | Lines 272-273: How does this compare to findings in the existing literature as were outlined in the introduction to this study? | We could not find a comparison between our finding and literature. In our study, distance from fire station is deemed less important than slope steepness and economic capacity, it is ranked as the third highest above saturated hydraulic conductivity, soil texture, and elevation (**P15L320-323**). However, our finding on slope steepness and economic capacity as the most important indicators is in agreement with previous studies. Roy et al. (2021) found that elevation and slope are the most important indicators while Moreira et al. (2021) found that social and economic indicators, such as population and income are the most important components for vulnerability.  In our study, population is categorized as exposure. We added a comparison in the revised manuscript (**P15L315-320**). |
| 26 | Lines 273-274: Would greater consensus have been reached by asking experts only to consider a smaller number of more relevant indicators? | We believe if the number of indicators is smaller, the expert will still weight the indicators based on the importance and thus, we will obtain the same rank but with different weights. |
| 27 | Lines 283-286: Are there existing studies regarding the influence of hydraulic conductivity which can provide further context for the decisions of the expert panel? | Yes. Versini et al. (2013) show an increase in river discharge after forest fire occurred in the Llobregat river basin, Spain (**P6L141-142**). Moreover, Seibert et al. (2010) and Folador et al. (2021) also show an increase in runoff after wildfire occurred (**P20L418-** |

| | | |
|---|---|---|
| | | **421**). We discussed this in Section 2.2 and 4.1. |
| 28 | Lines 296-297: Can you explain how decision fits within the context of understanding and mapping existing risk (prior to any other management interventions) and for predictions based upon assigned levels of societal intervention. | We agree with the experts that FVI should get more weight and then followed by the exposure and hazard. The flood risk could be better managed if the vulnerability is reduced by increasing societal interventions such as improving economic and institutional capacities and physical interventions such as slope, regreening, and soil improvement works. Hazard and exposure, on the other hand, are less manageable (**P16L354-357**). |
| 29 | Lines 301-303: What was the break-down of expertise in the panel? How significant was this difference in perspective? | We provided the expertise of the experts in the revised manuscript (**Supplementary Information Table S5** and **P11L224-226**) |
| 30 | Lines 306-307: What was the break-down of expertise in the panel? How significant was this difference in perspective? | The expert who indicates that exposure and vulnerability should be equally weighted has a background in social science and works on some technical aspects (**P17L366-368**). This shows that experts' perspective is influenced by their backgrounds and their working environment. |
| 31 | Lines 324-325: Can you clarify how notable this finding is? Or whether in fact this is just entirely due to the weightings assigned by the expert panel? | The burnt area is weighted for 30% in the flood hazard index (**P13L264**). Moreover, flood hazard index is only counted 20% in the calculation of flood risk compared to flood exposure index and flood vulnerability index (Eq. 7, **P16L360**). Thus, the effect of wildfires is greatly reduced in the flood risk calculation. We clarified this in the revised version (**P17L385-P18L388**). |
| 32 | Lines 342-343: Given the role of FHI, how much is the effect of wildfires controlled by the choice of weighting for burnt area: runoff? | For future flood risk analysis, the most influential factor is the FWI. Strong increase in FWI for RCP8.5 counteracts the decrease in runoff (**P18L403-P19L406**). |
| 33 | Lines 356-357: How much does it indicate this vs. indicating the perceived role cascading effect of wildfires given the role of the expert panel in determining the various weightings? | All the experts agreed that burnt area affects runoff and therefore most of them gave high weight to runoff (**P20L417-418**). |
| 34 | Lines 360-362: How possible would it be in future studies to further incorporate these previous findings to augment or replace the need for expert weightings? | A simple method to replace expert weightings is by using a correction factor. For example, if the runoff increases by a factor of 1.2 after wildfire occurred as found in Leopardi and Scorzini (2015), then the FHI is equal to 1.2 x runoff and FHI is equal to 1 x runoff for evaluating FHI without wildfire. We discussed this in the revised version (**P20L423-427**). |
| 35 | Lines 364-366: As discussed in Bedia et al 2013, does fuel moisture (and/or linked meteorological conditions) also play a role in limiting these large fire events? | This was discussed in Meyn et al. (2007) as cited by Bedia et al. (2013) (**P20L429-430**). We discussed the occurrence of wildfires that largely depends on forest (fuel) management, vegetation and land use practices, and fuel moisture in **P20L447-P21L448**. |
| 36 | Lines 378-379: As per previous comments, can you explicitly address the limitations of using FWI as a proxy for fire probability? | We discussed this limitation in **P22L500-507**. |

| 37 | Line 432: How did the annual timeframe affect the chosen FWI? Was this an average value for a whole year? Or a maximum value? | The FWI employed in the study is the mean fire weather index value over the European fire season (seasonal FWI, June-September) for 2050 and 2100 averaged from multi models. See Supplementary Table S2 for detailed information. |
|---|---|---|
| 38 | Section 4.4: Can you also discuss any limitations involved with the expert panel and the Analytical Hierarchy Process and associated data? | The limitation of AHP and expert panel was discussed in the revised version (**P22L509-512**). |
| 39 | Technical Corrections
Figure 2: Proofing comment - check figure quality/resolution as slightly blurred in places.
Line 149: Typesetting issue '1971 -2000'
Line 179: 'USDA' - Acronym needs to be defined on 1st use.
Line 312: Typo: 'into intervals of 16,7% per class'
Line 365: Typo: 'can be related to that the Mediterranean' | We thank for the feedback on Figure 2. We checked the quality of the figure.

The space was removed (**P7L165**).

We added the full name of USDA (**P8L196-197**).

We changed comma with dot (**P17L373**).

We rephrased the sentence accordingly (**P20L429-430**). |

**Additional References:**

Folador, L., Cislaghi, A., Vacchiano, G., and Masseroni, D.: Integrating Remote and In-Situ Data to Assess the Hydrological Response of a Post-Fire Watershed, Hydrology, 8(4), 169, https://doi.org/10.3390/hydrology8040169, 2021.

Leopardi, M. and Scorzini, A.: Effects of wildfires on peak discharges in watersheds [Technical Reports], iForest - Biogeosciences and Forestry, 8(3), 302–307, https://doi.org/10.3832ifor1120-007, 2015.

Mazdiyasni, O., and AghaKouchak, A.: Substantial increase in concurrent droughts and heatwaves in the United States, PNAS, 112(37), 11484-11489, https://doi.org/10.1073/pnas.1422945112, 2015.

He, B., Zhong, Z., Chen, D., Liu, J., Chen, Y., Miao, C., Ding, R., yuan, W., Guo, L., Huang, L., Hao, X., and Chen, A.: Lengthening dry spells intensify summer heatwaves, Geophysical Research Letters, 49, https://doi.org/10.1073/pnas.1422945112, 2022.

Tabari, H., Hosseinzadehtalaei, P., Thiery, W., and Willems, P.: Amplified Drought and Flood Risk Under Future Socioeconomic and Climatic Change, Earth's Future, 9(10), e2021EF002 295, https://doi.org/10.1029/2021EF002295, 2021.

Di Giuseppe, F., Rémy, S., Pappenberger, F., and Wetterhall, F.: Using the Fire Weather Index (FWI) to improve the estimation of fire emissions from fire radiative power (FRP) observations, Atmos. Chem. Phys., 18(8), 5359–5370, https://doi.org/10.5194/acp-18-5359-2018, 2018.

Roy, S., Bose, A., and Chowdhury, I. R.: Flood risk assessment using geospatial data and multi-criteria decision approach: a study from historically active flood-prone region of Himalayan foothill, India, Arabian Journal of Geosciences, 14(11), 999, https://doi.org/10.1007/s12517-021-07324-8, 2021.

McLennan, J. and Birch, A.: A potential crisis in wildfire emergency response capability? Australia's volunteer firefighters, Global Environ- mental Change Part B: Environmental Hazards, 6(2), 101–107, https://doi.org/10.1016/j.hazards.2005.10.003, 2005.

Seibert, J., McDonnell, J. J., and Woodsmith, R. D.: Effects of wildfire on catchment runoff response: a modelling approach to detect changes in snow-dominated forested catchments, Hydrology Research, 41(5), 378–390, https://doi.org/10.2166/nh.2010.036, 2010.

Versini, P. A., Velasco, M., Cabello, A., and Sempere-Torres, D.: Hydrological impact of forest fires and climate change in a Mediterranean basin, Natural Hazards, 66(2), 609–628, https://doi.org/10.1007/s11069-012-0503-z, 2013.

Leopardi, M. and Scorzini, A.: Effects of wildfires on peak discharges in watersheds [Technical Reports], iForest - Biogeosciences and Forestry, 8(3), 302–307, https://doi.org/10.3832ifor1120-007, 2015.

**Reply to reviewer 2**

We would like to thank the reviewer for valuable suggestions and comments. In this document, we reply to each of these. **L** refers to the line number. For example, **L65-70**, refers to lines 65-70.

| Reviewer 2 | | |
|---|---|---|
| 1 | This study shows an assessment of flood risk using a multi-criteria GIS-based approach incorporating wildfires and floods for the Ebro basin. The study tackles an important topic; it is a generally well written and well-argued paper, with a strong narrative and clear structure. | We would like to thank reviewer for the acknowledgement of the importance of the topic and the clarity of the paper. |
| 2 | At points in the paper, some terms are used somewhat interchangeably and not always defined. For example, in the title the authors use "effects", then later "impacts", then a mix. Similarly, what is meant by a cascade in this paper? Is it a cascade in terms of a trigger or something that increases risk (i.e. a mechanism or process that links these two hazards even if temporally), or a cascade in terms of impacts, or both? I feel as those terms are being used somewhat interchangeably in this paper.

 For example, in L12 in the abstract, the authors say "...especially when considering the cascading impacts of wildfires". I would question here what the cascade is? Or indeed, whether this is an impact? To me, this perhaps is more of knock on effect or something that increased the risk of something else through changing vulnerabilities (as the authors note later on) – a wildfire affects the flood risk through burnt area and so forth, which in turn may cause impacts for example – but is this process an impact? I would suggest that clearly defining these terms and then staying with them throughout would benefit the understanding for the reader. | We thank the reviewer for raising the terminology used in the paper. We agreed with the reviewer and decided to use the term effect instead of impact. Moreover, we provided the definition of cascading used in the manuscript, which is triggering or amplifying the flood risk (**P2L54-55**). We changed the impact term into effect throughout the revised manuscript. |
| 3 | The abstract mentions indicator only once, but a large part of the study is actually focused towards the integration of socio-economic indicators and land-use change information with 'conventional' hydrological properties and the cascading effects of wildfire to assess flood risk. This is a complicated endeavor within a multi-hazard/multi-risk approach, which is good to see, but I think the fact that this is approach should be made much clearer in the abstract and title so it is clear that the story is not solely about the cascading effects of wildfires and flood, it is more about better flood risk assessment as a whole, incorporating wildfires. | The referee has a valid point regarding mentioning indicators. In the revised version, we elaborated more in the indicators used in the study (**P1L3-5**). Moreover, we also expanded the abstract highlighting the multi-hazard/risk approach and how this could be done for better flood risk assessment.

 We thank the reviewer for the suggestion on the title and for stressing out the multi-hazard/risk instead of cascading. We revised the title into: "The cascading effect of wildfires on flood risk: A multi-hazard flood risk approach for Ebro River basin Spain (**P1L1**). |

| | | |
|---|---|---|
| | Later, in section 4.3 in the discussion, the authors state "Therefore, this research provides an example of how to integrate multiple hazards into risk evaluation by conducting comprehensive assessments that consider numerous drivers and indicators that will contribute to increased flood risk in the future." – this is a, I believe, a better (more correct?) framing for the study. I would suggest multi-hazard/risk be included in some way in the title, perhaps even removing the word cascade, and a reduction on the focus of the wildfire and flood cascades and more towards flood risk using multiple inputs from the start. | |
| 4 | The methods are comprehensive but section 2.2 is framed around flood risk indicators, however here this is where wildfire risk and indicators (FWI for example) is employed. Related to my point above, this section to me should be given a clear multi-hazard focus towards flood risk to make it clear that fires are part of this. Some subtle reframing and – importantly – including wildfire in the subtitle may be beneficial to guide the reader. | We thank the reviewer for the suggestion. We reframed the context of the article toward the focus of multi-hazard/risk (**P4L101-102**). We modified the heading of chapter 2.2 into "Indicators for flood risk assessment including wildfire risk" (**P4L100**). |
| 5 | The use and placement of the equations based on expert judgement FR, FEI etc is very confusing. FR is at the end of the methods section but if not defined until section 3.4. Then, additional equations, such as FVI, appear later on. Lines 213-4 says "This process allows us to calculate the Flood Hazard Index (FHI), the Flood Exposure Index (FEI), and the Flood Vulnerability Index (FVI), as denoted by Equation 1", however equation 1 shows the equation for FR. Equations 2 and 3 are not referenced from the text. Some terms, such as FS, are really hard to find the definitions of (one has to go looking in the text), and no units are provided. The use and presentation of these needs a rethink.

I would suggest that the equations are all placed in the methods and defined there, leaving the results to focus on the weighting by expert judgment and, therefore, the outcomes of the study. Indeed, many of the sub-sections 3.3 and 3.4 for example, stray into methods rather than results. Some careful reordering would really help the readability and accessibility. | The reviewer expresses concern about the use of equations based on the expert judgement and the structure of the results. Indeed, we place the results of FR in chapter 3.4 because FR cannot be calculated without knowing the FHI, FEI, and FVI values beforehand. This is the reason that we describe the results of FHI (chapter 3.1), FEI (3.2), and FVI (3.3) first. In the method, we present the main formula of calculating the FR, which consists of FHI, FEI, and FVI components. We provided all the flood risk component formulas in the chapter 2.5, including their full names and abbreviations (**P12L241-250**). In addition, we wrote a short paragraph in between chapters 3 and 3.1 explaining the structure of the results (**P13L259-262**). |
| 6 | Related to my above point, there is a large emphasis on expert judgement of the indicator weightings. It is not clear though quite how much emphasis they have on the results. Is seven people enough? Are they all | We thank the reviewer for the feedback. We provided the results of AHP in the Supplementary Information (E.g., **Figure S4, S7, S9, and S12**). Moreover, the background of the experts was provided in **P11L224-226** |

| | | |
|---|---|---|
| | from the Erbo region? Does this matter? The earlier phases of the study are quite analytical, but then the focus moves to a judgement based approach. Additional details on this process, perhaps in section 2.5, would in turn help understand and interpret the later results section. | and the **Supplementary Table S5**. Indeed, we only managed to interview seven experts although we sent interview requests more than seven. Some experts declined with various reasons and even one of them does not agree with the multi-risk framework that we formulated for the study. We discussed the limitation of only interviewing seven experts in the revised manuscript (**P22L509-512**). |
| 7 | The discussion is good and very readable. It provides some excellent additional information. I do think though that perhaps a bit more work may be needed to separate the location-specific findings based on local expert judgement and the wider findings that can be employed elsewhere. The authors don't really attempt to do this; instead the assumption is that the results shown here for Erbo would hold elsewhere. This localised expert judgement is not mentioned in the limitations in 4.4. The title at the start calls this paper a "study case" but really Erbo is the study here primarily. Making it clear that the findings for the Erbo maybe separate from broader interpretations would be beneficial in the discussion, including some detail on how this can be done (and the limitations of doing so), would really elevate this to be a usable example more broadly. | We thank the reviewer for the suggestion. Indeed, our study is only for the Ebro River basin. However, the approach employed in this study could be applied elsewhere as long as the indicators are the same. We added explanation about the applicability of our approach in Section 2.5 (**P12L251-257**).

The experts are not only coming from Spain but also other countries in Europe. The expert background was expanded in the revised manuscript (**P11L224-228** and **Supplementary Table S5**). |
| 8 | The conclusions in section 5 state "The research underscores the need for interdisciplinary collaboration…" in relation to the experts. While I agree with this statement, this is not a focus of the study and isn't presented up to this point. If the authors which to explore this narrative, this would be better placed in the discussion above. | The reviewer has a valid point and we moved the sentence into discussion (chapter 4.3) (**P22L482-483**). |
| 9 | Specific/minor comments:

Line 191-2      Please clarify or reword what is meant by future wildfire effects (FWI) and the burnt area? FWI is already defined, but what is meant by effects? | We thank the reviewer for spotting typo. Here we meant future wildfire danger indicated by FWI (**P8L210**). |

---

## Referee Report (RR1)

**REVIEW**

**The cascading effect of wildfires on flood risk: A multi-hazard flood risk approach for Ebro River basin Spain**
Samuel Jonson Sutanto, Matthijs Janssen, Mariana Madruga de Brito, and Maria del Pozo Garcia

The paper represents one of the first attempts at quantifying the wildfire-flood hazard interrelationship and evaluating its impacts including societal aspects. For this reason, I believe the contribution is scientifically valuable and deserves publication.

The authors effectively replied to the reviewers' comments and overall improved the manuscript. Nevertheless, **some minor but essential improvements are still required**, to better frame the work into the previous multi(-hazard)-risk literature.

1. The authors use extensively the expression "cascading", referring to, e.g., "the occurrence of cascading flooding after wildfires". At line 54, they explain that "cascading here means that the occurrence of wildfires preceding floods will trigger or amplify the risk of flooding".

   Nevertheless, floods are not directly triggered by wildfires, so it is not proper to talk about cascading. I suggest referring to "standardised" multi-hazard interaction mechanisms classifications available in the literature. What the authors are referring to is a typical case of "disposition alteration" as named by De Angeli et al. (2022), in which "there is no direct triggering of one hazard by another or any simultaneous temporal occurrence. Still, the occurrence of the first hazard can influence the frequency or the magnitude of the second one". This mechanism is also introduced by Tilloy et al. (2019) with the name "change condition".

   Ref:

   *De Angeli, S., Malamud, B. D., Rossi, L., Taylor, F. E., Trasforini, E., & Rudari, R. (2022). A multi-hazard framework for spatial-temporal impact analysis. International Journal of Disaster Risk Reduction, 73, 102829. Tilloy, A., Malamud, B. D., Winter, H., & Joly-Laugel, A. (2019). A review of quantification methodologies for multi-hazard interrelationships. Earth-Science Reviews, 196, 102881.*

2. The authors wrote that Versini et al. (2013) assessed flood risk, but then they affirmed that Versini et al. provided the hydrological probability of flooding, i.e. they did not assess risk but just

performed a probabilistic flood hazard assessment. I invite the authors to be careful to not mismatch hazard assessment and risk assessment.

3. If I understood well, the manuscript proposes advancements in three complementary directions:

   1) The modelling of the interaction mechanism between wildfire and flood, in terms of "disposition alteration" (see previous comment), for what concerns the hazard part
   2) The inclusion of socio-economic indicators, for what concerns the exposure and vulnerability dimensions
   3) The projection of future risk conditions

   These different aspects of novelty might be highlighted more clearly in the introduction, which is currently mixing all these concepts.

   Moreover, it is not so clear the innovation related to the second point. While the modelling of of the interaction mechanism between wildfire and flood covers a current gap, the inclusion of socio-economic indicators in flood risk assessment has been already largely explored in the literature. The authors should provide more indications about the innovation of this specific aspect. E.g., is it innovative because it has never been done in that specific case study area?

4. I feel a bit uncomfortable with the proposed "classification" of risk parameters into hazard, exposure and vulnerability. Indeed, some of the factors that the authors label as "vulnerability" are hazard parameters. I am referring, for example, to the Saturated Hydraulic Conductivity. More specifically, this is the flood hazard parameter which is "altered" by the wildfire, representing indeed the interaction mechanisms between the hazards that the authors introduced as a novel aspect. This multi-hazard mechanism is not well captured by the graphical representation of Fig. 2. This is also because the Saturated  Hydraulic Conductivity is seen as a vulnerability indicator rather than a hazard parameter.

---

## Author Response (AR2)

**Reply to reviewer 2**

In this document, P refers to the page number and L refers to the line number. For example, **P1L1-5** refers to page 1 lines 1-5.

| Reviewer 2 | | |
|---|---|---|
| 1 | The paper represents one of the first attempts at quantifying the wildfire- flood hazard interrelationship and evaluating its impacts including societal aspects. For this reason, I believe the contribution is scientifically valuable and deserves publication.
The authors effectively replied to the reviewers' comments and overall improved the manuscript. Nevertheless, some minor but essential improvements are still required, to better frame the work into the previous multi(-hazard)-risk literature. | We would like to thank reviewer for acknowledging the importance of the topic and recognizing the contribution of our paper. We also appreciate the valuable feedback provided to improve the manuscript. |
| 2 | The authors use extensively the expression "cascading", referring to, e.g., "the occurrence of cascading flooding after wildfires". At line 54, they explain that "cascading here means that the occurrence of wildfires preceding floods will trigger or amplify the risk of flooding".

Nevertheless, floods are not directly triggered by wildfires, so it is not proper to talk about cascading. I suggest referring to "standardised" multi-hazard interaction mechanisms classifications available in the literature. What the authors are referring to is a typical case of "disposition alteration" as named by De Angeli et al. (2022), in which "there is no direct triggering of one hazard by another or any simultaneous temporal occurrence. Still, the occurrence of the first hazard can influence the frequency or the magnitude of the second one". This mechanism is also introduced by Tilloy et al. (2019) with the name "change condition".

Ref:
De Angeli, S., Malamud, B. D., Rossi, L., Taylor, F. E., Trasforini, E., & Rudari, R. (2022). A multi-hazard framework for spatial-temporal impact analysis. International Journal of Disaster Risk Reduction, 73, 102829.
Tilloy, A., Malamud, B. D., Winter, H., & Joly-Laugel, A. (2019). A review of quantification methodologies for multi-hazard interrelationships. Earth- Science Reviews, 196, 102881. | We thank the reviewer for raising the concern with the terminology "cascading" used in the paper. In the previous version, we would like to emphasize that the burned area will amplify the risk of flooding, which is true. However, the reviewer made a valid point based on the provided literature. We agreed with the reviewer that wildfires alter the disposition of flood hazard by changing soil characteristics. Therefore, we decided to replace the term "cascading" with "amplify" or "effect" (E.g., **P1L3**). We also explained the interaction between wildfires and floods in the revised manuscript (**P2L54-55**) and highlighted this interaction in Figure 2 (**P6**). |

| 3 | The authors wrote that Versini et al. (2013) assessed flood risk, but then they affirmed that Versini et al. provided the hydrological probability of flooding, i.e. they did not assess risk but just performed a probabilistic flood hazard assessment. I invite the authors to be careful to not mismatch hazard assessment and risk assessment. | We thank the reviewer for the careful reading. We changed the word from flood risk to flood occurrence and probability (**P2L47-48**). |
|---|---|---|
| 4 | If I understood well, the manuscript proposes advancements in three complementary directions:

1) The modelling of the interaction mechanism between wildfire and flood, in terms of "disposition alteration" (see previous comment), for what concerns the hazard part
2) The inclusion of socio-economic indicators, for what concerns the exposure and vulnerability dimensions
3) The projection of future risk conditions

These different aspects of novelty might be highlighted more clearly in the introduction, which is currently mixing all these concepts.

Moreover, it is not so clear the innovation related to the second point. While the modelling of the interaction mechanism between wildfire and flood covers a current gap, the inclusion of socio- economic indicators in flood risk assessment has been already largely explored in the literature. The authors should provide more indications about the innovation of this specific aspect. E.g., is it innovative because it has never been done in that specific case study area? | We thank the reviewer for the suggestion. We highlighted these three aspects in the revised version (**P2L56-P3L65**).

Our study is innovative because it includes comprehensive socio-economic indicators for flood risk assessment (see Figure 2). For the exposure component, we include population, economic values of the regions, and road infrastructure. Many studies only consider population or land use as the main exposure component (Foudi et al., 2015; Gain et al., 2015). For the vulnerability component, we consider both physical and social factors, such as topography, land cover, soil infiltration capacity, economic capacity, and institutional capacity. The use of these wide-ranging vulnerability indicators in the flood risk assessment highlights the novelty of our study, which has not been included in many studies (Brouwer et al., 2007; Gain et al., 2015; Cai et al., 2019). We have further developed this advancement in the revised manuscript (**P3L60-64**). |
| 5 | I feel a bit uncomfortable with the proposed "classification" of risk parameters into hazard, exposure and vulnerability. Indeed, some of the factors that the authors label as "vulnerability" are hazard parameters. I am referring, for example, to the Saturated Hydraulic Conductivity. More specifically, this is the flood hazard parameter which is "altered" by the wildfire, representing indeed the interaction mechanisms between the hazards that the authors introduced as a novel aspect. This multi-hazard mechanism is not well captured by the graphical representation of Fig. 2. This is also because the Saturated Hydraulic Conductivity is seen as a vulnerability indicator rather than a hazard parameter. | The reviewer expresses concern about the placement of certain parameters under vulnerability, exposure, and hazard, specifically the soil hydraulic conductivity as one of the vulnerability components and not as a hazard parameter. In our study, we aim to make a clear distinction between hazard and consequences, which consist of exposure and vulnerability. We did not classify hydraulic conductivity as a hazard because it is not a hazard itself. This parameter is altered by the hazard, here is wildfire. Wildfires make the area more vulnerable to flooding due to reduced infiltration capacity (**P7L144-146**). Based on this reasoning, we categorized saturated hydraulic conductivity as one of the vulnerability components.
In Figure 2, we drawn arrows to link the effects of wildfires for current and future scenarios on saturated hydraulic conductivity. In the revised |

| | | version, we modified the Figure 2 by drawing a red arrow from saturated hydraulic conductivity to runoff coefficient and providing an explanation that wildfires increase the runoff coefficient (**P6**). |
|---|---|---|

**References**

De Angeli, S., Malamud, B. D., Rossi, L., Taylor, F. E., Trasforini, E., and Rudari, R.: A multi-hazard framework for spatial-temporal impact analysis, International Journal of Disaster Risk Reduction, 73, 102829, 2022.

Tilloy, A., Malamud, B. D., Winter, H., and Joly-Laugel, A.: A review of quantification methodologies for multi-hazard interrelationships, Earth- Science Reviews, 196, 102881, 2019.

Foudi, S., Osés-Eraso, N., and Tamayo, I.: Integrated spatial flood risk assessment: The case of Zaragoza, Land Use Policy, 42, 278-292, https://doi.org/10.1016/j.landusepol.2014.08.002, 2015.

Gain, A. K., Mojtahed, V., Biscaro, C., Balbi, S., and Giupponi, C.: An integrated approach of flood risk assessment in the eastern part of Dhaka City, Natural Hazards, 79, 1499-1530, https://doi.org/10.1007/s11069-015-1911-7, 2015.

Brouwer, R., Akter, S., Brander, L., and Haque, E.: Socioeconomic Vulnerability and Adaptation to Environmental Risk: A Case Study of Climate Change and Flooding in Bangladesh, Risk Analysis, 27(2), 313–326, https://doi.org/10.1111/j.1539-6924.2007.00884.x, 2007.

Cai, T., Li, X., Ding, X., Wang, J., and Zhan, J.: Flood risk assessment based on hydrodynamic model and fuzzy comprehensive evaluation with GIS technique, International Journal of Disaster Risk Reduction, 35, 101 077, https://doi.org/10.1016/j.ijdrr.2019.101077, 2019.